# Autoencoding Reduced Order Models for Control through the Lens of Dynamic Mode Decomposition

## Abstract

Modeling and control of high-dimensional dynamical systems often involve some dimensionality reduction techniques to construct a lower-order model that makes the associated task computationally feasible or less demanding. In recent years, two techniques have become widely popular for analysis and reduced order modeling of high-dimensional dynamical systems: (1) dynamic mode decomposition and (2) deep autoencoding learning. This paper establishes a connection between dynamic mode decomposition and autoencoding learning for controlled dynamical systems. Specifically, we first show that an optimization objective for learning a linear autoencoding reduced order model can be formulated such that its solution closely resembles the solution obtained by the *dynamic mode decomposition with control* algorithm. The linear autoencoding architecture is then extended to a deep autoencoding architecture to learn a nonlinear reduced order model. Finally, the learned reduced order model is used to design a controller utilizing stability-constrained deep neural networks. The studied framework does not require knowledge of the governing equations of the underlying system and learns the model and controller solely from time series data of observations and actuations. We provide empirical analysis on modeling and control of spatiotemporal high-dimensional systems, including fluid flow control.

## 1 Introduction

Designing controllers for high-dimensional dynamical systems remains a challenge as many control algorithms become computationally prohibitive in high dimensions. Typically, a *reduce-then-design* approach (Atwell et al. (2001)) is used in practice, which involves two steps: (1) develop a reduced order model (ROM) using dimensionality reduction techniques and (2) design a controller for that reduced order model (Figure 1a). For controlled dynamical systems, the reduced order modeling approaches either combine analytical techniques with empirical approximation (Willcox & Peraire (2002)) or are purely data-driven (Juang & Pappa (1985); Juang et al. (1993); Proctor et al. (2016)). Among these, the dynamic mode decomposition (DMD) based methods have become widely popular in recent years due to a strong connection between DMD and Koopman operator theory (Rowley et al. (2009)). Another recent research trend involves utilizing deep neural networks (DNNs), particularly autoencoders, for modeling and control of high-dimensional dynamical systems (Lusch et al. (2018); Erichson et al. (2019); Eivazi et al. (2020); Morton et al. (2018); Bounou et al. (2021); Chen et al. (2021); Ping et al. (2021)).

In this paper, we provide a perspective connecting DMD and autoencoding reduced order models for controlled dynamical systems and present a framework to learn control policies for such systems by means of the DNN-based reduced order models. We first formulate an objective function for data-driven learning of controlled dynamical systems in a linear autoencoding configuration. We analytically show that the associated objective function encourages a linear ROM that closely resembles the lower-order model obtained using the *dynamic mode decomposition with control* (DMDc) algorithm (Proctor et al. (2016)). The linear autoencoding architecture is designed in such a way that its components can be replaced with DNNs and the corresponding objective function can be optimized by gradient descent to obtain a nonlinear ROM. The architecture with DNN components, DeepROM, closely resembles the aforementioned deep autoencoding architectures used in recent literature for the prediction and control of dynamical systems. The learned

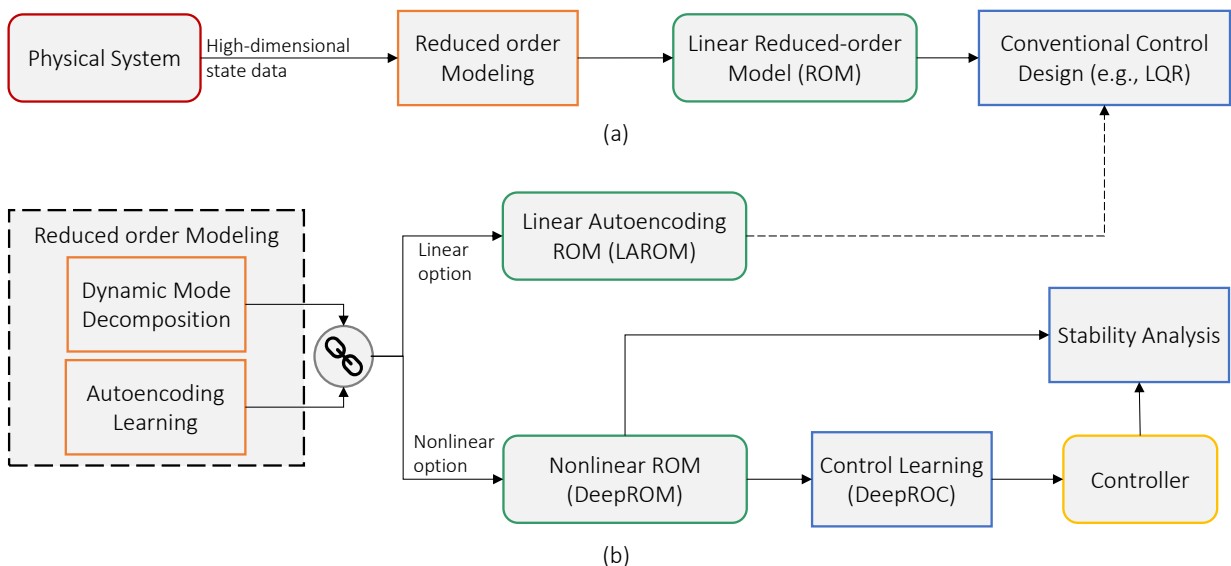

Figure 1: (a) Reduce-then-design paradigm for designing control for high-dimensional systems, (b): our work in the context. The rounded-corner rectangles denote the actual physical system (red outlined) or its models (green outlined) or a controller (yellow outlined). The sharp-corner rectangles indicate the techniques/procedure to obtain the models or controllers. Among those sharp-corner rectangles, the ones outlined in orange are associated with modeling, while the ones outlined in blue are associated with control. This work proposes an autoencoding learning framework that establishes a *link* with dynamic mode decomposition. The similarity with dynamic mode decomposition for control is shown through a linear autoencoding reduced order model while the prediction and control performance is evaluated using a deep autoencoding reduced order model. The dashed arrow in the figure represents the potential control methods that can be applied to the linear ROM.

DNN-based reduced order model is then used in a control learning framework, deep reduced order control (DeepROC), to design a controller for the underlying system. The control policy is learned by jointly training two DNNs: one stability-constrained DNN predicts a target closed-loop dynamics for the learned ROM while the other DNN serves as a controller to achieve that target dynamics. We analytically show that keeping the joint learning objective within a sufficiently small value implies stability for the closed-loop ROM in terms of *ultimate boundedness*, i.e., trajectories starting close to the desired state stay close to the desired state. The overall workflow of this paper is shown in Figure 1(b). We provide empirical analysis using examples of spatiotemporal physical processes such as reaction–diffusion and fluid flow systems. In summary, our contributions are as follows:

- For controlled dynamical systems, we show that an objective function can be formulated in a linear autoencoding configuration and optimized by gradient descent such that the corresponding linear ROM closely resembles the ROM obtained using the DMDc algorithm.

- We extend the linear autoencoding configuration to a deep autoencoding configuration to learn a DNN-based nonlinear ROM.

- We analytically show that a DNN controller can be trained such that the closed-loop trajectories of the learned ROM remain ultimately bounded.

- We empirically show the similarity of the linear autoencoding reduced order model (LAROM) with DMDc and evaluate the prediction performance of DeepROM and control performance of DeepROC in experiments with high-dimensional systems, including suppressing vortex shedding in fluid flow.

## 2 Related Work

In recent years, deep learning has seen widespread application in scientific and engineering problems, including understanding complex dynamics of large-scale or high-dimensional systems and solving associated computational tasks. The majority of the research in this area focuses on the modeling and prediction of such complex dynamics using deep neural networks (DNNs) (Xingjian et al. (2015); Long et al. (2018); Raissi (2018); Seo et al. (2019); Ayed et al. (2019); Donà et al. (2020)) and has found application in several fields including fluid flow (Erichson et al. (2019); Eivazi et al. (2020); Srinivasan et al. (2019)), biochemical and electric power systems (Yeung et al. (2019)), climate and ocean systems (Scher (2018); Ren et al. (2021); Yang et al. (2017); De Bézenac et al. (2019)), and structural analysis Zhang et al. (2020), just to name a few.

A second line of research, though relatively less prevalent than modeling and prediction, is utilizing deep learning for controlling high-dimensional systems and aligns closely with our work. Most of these works focus on fluid flow control tasks. Rabault et al. (2019); Tang et al. (2020) applied deep reinforcement learning in active flow control for vortex shedding suppression and used a system-specific reward function involving lift and drag. Ma et al. (2018) used an autoencoder for encoding high-dimensional fluid state to low-dimensional features and trained RL agents with those features to control rigid bodies in a coupled 2D system involving both fluid and rigid bodies. Garnier et al. (2021) also used an autoencoder for feature extraction to train an RL agent for controlling the position of a small cylinder in a two-cylinder fluid system to reduce the total drag. Beintema et al. (2020) used deep reinforcement learning with system-specific rewards to control a Rayleigh–Bénard system with the aim of reducing convective effects. Model-free deep reinforcement learning methods have high sample complexity necessitating a large number of interactions with the environment and often require system-specific reward construction. In contrast, we consider learning the control policies offline with limited pre-collected data. Moreover, we utilize standard distance-based metrics with respect to a hypothesized target dynamics instead of relying on any system-specific rewards or loss functions. Model-free RL methods require running numerical solvers in every iteration to provide feedback to the agents, which is computationally expensive. The same concern arises for the methods involving differentiable simulators. For example, Holl et al. (2020) used a differentiable partial differential equation (PDE) solver to generate gradient-based feedback for a predictor-corrector scheme to control PDE-driven high-dimensional systems. Takahashi et al. (2021) too integrated a differentiable simulator with DNNs to learn control in coupled solid-fluid systems. In comparison, our method avoids the need for computationally intensive simulators during the learning process as it learns from pre-collected data in an offline manner.

The alternative to model-free methods takes the traditional approach: develop a model first and then use that to design controllers. Deep learning is now being used in this process by developing frameworks like DeepMPC (Lenz et al. (2015)) which incorporates DNN features in model predictive control (MPC). Bieker et al. (2020) and Morton et al. (2018) utilized the DeepMPC framework for fluid flow control. Bieker et al. (2020) used a recurrent neural network to model the dynamics of only control-relevant quantities (i.e. lift and drag) of the system, which is then employed in an MPC framework for the flow control tasks. Morton et al. (2018) followed the method proposed by Takeishi et al. (2017) and used DNN-based embedding to first learn a linear reduced order model in Koopman invariant subspace and then incorporate it in the MPC framework. Similar approaches have been applied to other high-dimensional control tasks like control from video input (Bounou et al. (2021)), automatic generation control in wind farms in the presence of dynamic wake effect (Chen et al. (2021)), and transient stabilization in power grids (Ping et al. (2021)). These model-based methods constrain the latent dynamic models to be linear that work well within a short time window. Khodkar et al. (2019) showed that the linear combination of a finite number of dynamic modes may not accurately represent the long-term nonlinear characteristics of complex dynamics and adding nonlinear forcing terms yields better prediction accuracy (Eivazi et al. (2020)). The linear ROMs are needed to be updated with online observations during operation for better prediction accuracy. Accordingly, the aforementioned model-based approaches utilize the MPC framework to optimize the control policy online using the updated dynamic model. Running online optimization at every step may not be computationally feasible in many scenarios. Conversely, we investigate if a nonlinear ROM provides a more accurate prediction over a longer time window so that an offline control learning method can be used.

## 3 Problem and Preliminaries

### 3.1 Problem statement

Consider a time-invariant controlled dynamical system

$$\frac{d\boldsymbol{x}}{dt} = f(\boldsymbol{x}, \boldsymbol{u}), \tag{1}$$

where $\boldsymbol{x}(t) \in \mathbb{X} \subset \mathbb{R}^{d_{\boldsymbol{x}}}, d_{\boldsymbol{x}} >> 1$ and $\boldsymbol{u}(t) \in \mathbb{U} \subset \mathbb{R}^{d_{\boldsymbol{u}}}$ are the system state and the actuation (or control input), respectively, at time $t$. Our objective is to learn a feedback controller $\boldsymbol{u} = \pi(\boldsymbol{x})$ for this high-dimensional ($d_{\boldsymbol{x}} >> 1$) system of (1) to stabilize it at a desired state in a data-driven reduce-then-design approach when the nonlinear function $f : \mathbb{X} \times \mathbb{U} \to \mathbb{R}^{d_{\boldsymbol{x}}}$ is unknown. We assume that we have observations from the system consisting of time series data $\boldsymbol{x}(t_i), i = 0, 1, \cdots, n$ subjected to random values of actuations $\boldsymbol{u}(t_i), i = 0, 1, \cdots, (n-1)$.

Note, we use $v$ (in place of $v(t)$ for brevity) as notation for any continuous-time variable (e.g., system state, control input), whereas $v(t_i)$ is used to denote their discrete sample at time instance $t_i$.

### 3.2 Stabilization of controlled systems

We assume that the system we are aiming to stabilize at an equilibrium point is *locally stabilizable*. Suppose the function $f$ in (1) is locally Lipschitz and ($\boldsymbol{x} = \boldsymbol{0}, \boldsymbol{u} = \boldsymbol{0}$) is an equilibrium pair of the system, i.e., $f(\boldsymbol{0}, \boldsymbol{0}) = \boldsymbol{0}$. The system (1) is said to be *locally stabilizable* with respect to the equilibrium pair if there exists a locally Lipschitz function

$$\pi : \mathbb{X}_0 \to \mathbb{U}, \quad \pi(\boldsymbol{0}) = \boldsymbol{0},$$

defined on some neighborhood $\mathbb{X}_0 \subset \mathbb{X}$ of the origin $\boldsymbol{x} = \boldsymbol{0}$ for which the closed-loop system

$$\frac{d\boldsymbol{x}}{dt} = f(\boldsymbol{x}, \pi(\boldsymbol{x})) \tag{2}$$

is locally *asymptotically stable*, i.e. $\|\boldsymbol{x}(t_0)\| < \delta$ implies $\lim_{t \to \infty} \boldsymbol{x}(t) = \boldsymbol{0}$ (Sontag (2013)). We discuss the criteria for stability and asymptotic stability in the following paragraph.

Stability of the closed-loop system $\frac{d\boldsymbol{x}}{dt} = f(\boldsymbol{x}, \pi(\boldsymbol{x})) = h(\boldsymbol{x})$ at equilibrium points can be analyzed using the method of Lyapunov. Let $\mathcal{V} : \mathbb{X} \to \mathbb{R}$ be a continuously differentiable function such that

$$\mathcal{V}(\boldsymbol{0}) = 0, \quad \text{and} \quad \mathcal{V}(\boldsymbol{x}) > 0 \quad \forall \ \boldsymbol{x} \in \mathbb{X} \setminus \{\boldsymbol{0}\}, \tag{3}$$

and the time derivative of $\mathcal{V}$ along the trajectories

$$\frac{d\mathcal{V}}{dt} = \nabla\mathcal{V}(\boldsymbol{x})^\top \frac{d\boldsymbol{x}}{dt} = \nabla\mathcal{V}(\boldsymbol{x})^\top h(\boldsymbol{x}) \leq 0 \quad \forall \ \boldsymbol{x} \in \mathbb{X}. \tag{4}$$

Then, the equilibrium point $\boldsymbol{x} = \boldsymbol{0}$ is *stable*, i.e., for each $\epsilon > 0$, there exists a $\delta = \delta(\epsilon) > 0$ such that $\|\boldsymbol{x}(t_0)\| < \delta$ implies $\|\boldsymbol{x}(t)\| < \epsilon, \ \forall t > t_0$. The function $\mathcal{V}$ with the above properties is called a Lyapunov function. If $\frac{d\mathcal{V}}{dt} < 0$ in some subset $\mathbb{X}_s \subset \mathbb{X} \setminus \{\boldsymbol{0}\}$, then $\boldsymbol{x} = \boldsymbol{0}$ is *locally* asymptotically stable. Moreover, if there exist positive constants $c_1, c_2, c_3$ and $c_4$ such that

$$c_1\|\boldsymbol{x}\|^2 \leq \mathcal{V}(\boldsymbol{x}) \leq c_2\|\boldsymbol{x}\|^2, \tag{5}$$

and

$$\nabla\mathcal{V}(\boldsymbol{x})^\top h(\boldsymbol{x}) \leq -c_3\|\boldsymbol{x}\|^2, \quad \forall \ \boldsymbol{x} \in \mathbb{X}_s, \tag{6}$$

then $\boldsymbol{x} = \boldsymbol{0}$ is *exponentially stable*, i.e., there exist positive constants $\delta, \lambda$ and $\gamma$ such that $\|\boldsymbol{x}(t)\| \leq \lambda\|\boldsymbol{x}(t_0)\|e^{-\gamma(t-t_0)}, \ \forall\|\boldsymbol{x}(t_0)\| < \delta$ (Khalil (2002)).

In this paper, we assume that the system we are aiming to stabilize at an equilibrium point is stabilizable in the sense of the aforementioned definition and criteria, i.e., there exists a continuously differentiable function

$\mathcal{V}$ and a Lipschitz continuous control law $\pi$ such that criteria (3) and (4) are conformed. To ensure the stability of the closed-loop reduced order model at equilibrium, we utilize a target dynamics hypothesis that is exponentially stable at the origin, i.e., (5) and (6) are satisfied as well for this target dynamics. Detail on the target dynamics hypothesis is discussed in subsection 4.2.

Though the above formulation is for stabilization at an equilibrium point $\boldsymbol{x} = \boldsymbol{0}$, the same can be used to stabilize the system at any arbitrary point $\boldsymbol{x}_{ss}$. In that case, a steady-state control input $\boldsymbol{u}_{ss}$ is required that can maintain the equilibrium at $\boldsymbol{x}_{ss}$, i.e., $f(\boldsymbol{x}_{ss}, \boldsymbol{u}_{ss}) = \boldsymbol{0}$. The change of variables $\boldsymbol{x}_e = \boldsymbol{x} - \boldsymbol{x}_{ss}, \boldsymbol{u}_e = \boldsymbol{u} - \boldsymbol{u}_{ss}$ leads to a transformed system where we can apply the aforementioned formulation of stabilization. The overall control, in this case, $\boldsymbol{u} = \boldsymbol{u}_e + \boldsymbol{u}_{ss}$ comprises a feedback component $\boldsymbol{u}_e$ and a feedforward component $\boldsymbol{u}_{ss}$ (Khalil (2002)).

### 3.3 Dynamic mode decomposition with control

DMD (Schmid (2010)) is a data-driven method that reconstructs the underlying dynamics using only a time series of snapshots from the system. DMD computes a modal decomposition where each mode is associated with an oscillation frequency and decay/growth rate. DMD has become a widely used technique for spectral analysis of dynamical systems. DMDc (Proctor et al. (2016)) is an extension of DMD for dynamical systems with control. DMDc seeks best-fit linear operators $\boldsymbol{A}$ and $\boldsymbol{B}$ between successive observed states and the actuations:

$$\hat{\boldsymbol{x}}(t_{i+1}) = \boldsymbol{A}\boldsymbol{x}(t_i) + \boldsymbol{B}\boldsymbol{u}(t_i), \quad i = 0, 1, \cdots, n-1, \tag{7}$$

where $\hat{\boldsymbol{x}}(t)$ denotes an approximation of $\boldsymbol{x}(t)$, $\boldsymbol{A} \in \mathbb{R}^{d_{\boldsymbol{x}} \times d_{\boldsymbol{x}}}$, and $\boldsymbol{B} \in \mathbb{R}^{d_{\boldsymbol{x}} \times d_{\boldsymbol{u}}}$. Direct analysis of (7) could be computationally prohibitive for $d_{\boldsymbol{x}} >> 1$. DMDc leverages dimensionality reduction to compute a ROM

$$\boldsymbol{x}_{\mathrm{R,DMDc}}(t_i) = \boldsymbol{E}_{\mathrm{DMDc}}\boldsymbol{x}(t_i), \tag{8a}$$

$$\boldsymbol{x}_{\mathrm{R,DMDc}}(t_{i+1}) = \boldsymbol{A}_{\mathrm{R,DMDc}}\boldsymbol{x}_{\mathrm{R,DMDc}}(t_i) + \boldsymbol{B}_{\mathrm{R,DMDc}}\boldsymbol{u}(t_i), \quad i = 0, 1, \cdots, n-1, \tag{8b}$$

which retains the dominant dynamic modes of (7). Here, $\boldsymbol{x}_{\mathrm{R,DMDc}}(t_i) \in \mathbb{R}^{r_{\boldsymbol{x}}}$ is the reduced state, where $r_{\boldsymbol{x}} << d_{\boldsymbol{x}}$, and $\boldsymbol{E}_{\mathrm{DMDc}} \in \mathbb{R}^{r_{\boldsymbol{x}} \times d_{\boldsymbol{x}}}$, $\boldsymbol{A}_{\mathrm{R,DMDc}} \in \mathbb{R}^{r_{\boldsymbol{x}} \times r_{\boldsymbol{x}}}, \boldsymbol{B}_{\mathrm{R,DMDc}} \in \mathbb{R}^{r_{\boldsymbol{x}} \times d_{\boldsymbol{u}}}$. The full state is reconstructed from the reduced state using the transformation $\hat{\boldsymbol{x}}(t_i) = \boldsymbol{D}_{\mathrm{DMDc}}\boldsymbol{x}_{\mathrm{R,DMDc}}(t_i)$, where $\boldsymbol{D}_{\mathrm{DMDc}} \in \mathbb{R}^{d_{\boldsymbol{x}} \times r_{\boldsymbol{x}}}$. DMDc computes truncated singular value decomposition (SVD) of the data matrices $\boldsymbol{Y} = [\boldsymbol{x}(t_1), \boldsymbol{x}(t_2), \cdots, \boldsymbol{x}(t_n)] \in \mathbb{R}^{d_{\boldsymbol{x}} \times n}$ and $\boldsymbol{\Omega} = [\boldsymbol{\omega}(t_0), \boldsymbol{\omega}(t_1), \cdots, \boldsymbol{\omega}(t_{n-1})] \in \mathbb{R}^{(d_{\boldsymbol{x}}+d_{\boldsymbol{u}}) \times n}$, $\boldsymbol{\omega}(t_i) = [\boldsymbol{x}(t_i)^{\top}, \boldsymbol{u}(t_i)^{\top}]^{\top} \in \mathbb{R}^{d_{\boldsymbol{x}}+d_{\boldsymbol{u}}}$ as follows:

$$\boldsymbol{Y} = \widehat{\boldsymbol{U}}_{\boldsymbol{Y}} \widehat{\boldsymbol{\Sigma}}_{\boldsymbol{Y}} \widehat{\boldsymbol{V}}_{\boldsymbol{Y}}^{\top}, \quad \boldsymbol{\Omega} = \widehat{\boldsymbol{U}}_{\boldsymbol{\Omega}} \widehat{\boldsymbol{\Sigma}}_{\boldsymbol{\Omega}} \widehat{\boldsymbol{V}}_{\boldsymbol{\Omega}}^{\top}, \tag{9}$$

where $\widehat{\boldsymbol{U}}_{\boldsymbol{Y}} \in \mathbb{R}^{d_{\boldsymbol{x}} \times r_{\boldsymbol{x}}}, \widehat{\boldsymbol{\Sigma}}_{\boldsymbol{Y}} \in \mathbb{R}^{r_{\boldsymbol{x}} \times r_{\boldsymbol{x}}}, \widehat{\boldsymbol{V}}_{\boldsymbol{Y}} \in \mathbb{R}^{n \times r_{\boldsymbol{x}}}, \widehat{\boldsymbol{U}}_{\boldsymbol{\Omega}} \in \mathbb{R}^{(d_{\boldsymbol{x}}+d_{\boldsymbol{u}}) \times r_{\boldsymbol{x}\boldsymbol{u}}}, \widehat{\boldsymbol{\Sigma}}_{\boldsymbol{\Omega}} \in \mathbb{R}^{r_{\boldsymbol{x}\boldsymbol{u}} \times r_{\boldsymbol{x}\boldsymbol{u}}}$, and $\widehat{\boldsymbol{V}}_{\boldsymbol{\Omega}} \in \mathbb{R}^{n \times r_{\boldsymbol{x}\boldsymbol{u}}}$. $r_{\boldsymbol{x}} < \min(d_{\boldsymbol{x}}, n)$ and $r_{\boldsymbol{x}} < r_{\boldsymbol{x}\boldsymbol{u}} < \min(d_{\boldsymbol{x}} + d_{\boldsymbol{u}}, n)$ denote the truncation dimensions of SVDs. Utilizing the SVDs of (9) the parameters of the ROM (8) is obtained as

$$\boldsymbol{E}_{\mathrm{DMDc}} = \widehat{\boldsymbol{U}}_{\boldsymbol{Y}}^{\top}, \quad \boldsymbol{D}_{\mathrm{DMDc}} = \widehat{\boldsymbol{U}}_{\boldsymbol{Y}}, \tag{10a}$$

$$\boldsymbol{A}_{\mathrm{R,DMDc}} = \widehat{\boldsymbol{U}}_{\boldsymbol{Y}}^{\top} \boldsymbol{Y} \widehat{\boldsymbol{V}}_{\boldsymbol{\Omega}} \widehat{\boldsymbol{\Sigma}}_{\boldsymbol{\Omega}}^{-1} \widehat{\boldsymbol{U}}_{\boldsymbol{\Omega},1}^{\top} \widehat{\boldsymbol{U}}_{\boldsymbol{Y}}, \quad \boldsymbol{B}_{\mathrm{R,DMDc}} = \widehat{\boldsymbol{U}}_{\boldsymbol{Y}}^{\top} \boldsymbol{Y} \widehat{\boldsymbol{V}}_{\boldsymbol{\Omega}} \widehat{\boldsymbol{\Sigma}}_{\boldsymbol{\Omega}}^{-1} \widehat{\boldsymbol{U}}_{\boldsymbol{\Omega},2}^{\top}, \tag{10b}$$

where $\widehat{\boldsymbol{U}}_{\boldsymbol{\Omega},1} \in \mathbb{R}^{d_{\boldsymbol{x}} \times r_{\boldsymbol{x}\boldsymbol{u}}}, \widehat{\boldsymbol{U}}_{\boldsymbol{\Omega},2} \in \mathbb{R}^{d_{\boldsymbol{u}} \times r_{\boldsymbol{x}\boldsymbol{u}}}$, and $\widehat{\boldsymbol{U}}_{\boldsymbol{\Omega}}^{\top} = [\widehat{\boldsymbol{U}}_{\boldsymbol{\Omega},1}^{\top} \quad \widehat{\boldsymbol{U}}_{\boldsymbol{\Omega},2}^{\top}]$.

## 4 Method

As mentioned earlier, in the reduce-then-design approach, we first need to develop a ROM and then design a controller using that ROM. A controller designed for the ROM is expected to perform well in the full system only if the ROM effectively captures the dynamic characteristics of the underlying system. In this section, we first describe how to design a ROM that effectively captures the relation between successive observations and actuation. Next, we delineate the process for learning controllers utilizing the learned ROM.

### 4.1 Learning a reduced order model

DMDc can extract the dominant modes of underlying dynamics in a reduced order model (Proctor et al. (2016)). In order to develop a nonlinear ROM utilizing DNNs that effectively capture the underlying dynamics, we first investigate if we can obtain a linear ROM similar to DMDc, in a gradient descent arrangement. Specifically, we analyze optimization objectives that encourage a DMDc-like solution for a reduced order modeling problem using linear networks (single layer without nonlinear activation). Consider the following reduced order modeling problem

$$\boldsymbol{x}_{\mathrm{R}}(t_i) = \boldsymbol{E_x}\boldsymbol{x}(t_i), \ \boldsymbol{x}_{\mathrm{R}}(t_{i+1}) = \boldsymbol{A}_{\mathrm{R}}\boldsymbol{x}_{\mathrm{R}}(t_i) + \boldsymbol{B}_{\mathrm{R}}\boldsymbol{u}(t_i), \hat{\boldsymbol{x}}(t_i) = \boldsymbol{D_x}\boldsymbol{x}_{\mathrm{R}}(t_i), \ i = 0, 1, \cdots, n-1, \quad (11)$$

where the linear operators $\boldsymbol{E_x} \in \mathbb{R}^{r_x \times d_x}$ and $\boldsymbol{D_x} \in \mathbb{R}^{d_x \times r_x}$ projects and reconstructs back, respectively, the high-dimensional system state to and from a low-dimensional feature $\boldsymbol{x}_{\mathrm{R}} \in \mathbb{R}^{r_x}$. The linear operators $\boldsymbol{A}_{\mathrm{R}} \in \mathbb{R}^{r_x \times r_x}$ and $\boldsymbol{B}_{\mathrm{R}} \in \mathbb{R}^{r_x \times d_u}$ describe the relations between successive reduced states and actuations. We refer to this reduced order model with linear networks as linear autoencoding ROM or LAROM. In the following, we first analyze the solution of the optimization objective of LAROM for a fixed *encoder* $\boldsymbol{E_x}$. Then we establish a connection between the solution of LAROM and the solution of DMDc, and further discuss the choice of the encoder to promote similarity between the two. Finally, we extend the linear model to a DNN-based model, which we refer to as DeepROM.

#### 4.1.1 Analysis of the linear reduced order model for a fixed encoder

The DMDc algorithm essentially solves for $\widetilde{\boldsymbol{G}} \in \mathbb{R}^{r_x \times (d_x + d_u)}$ to minimize $\frac{1}{n} \sum_{i=0}^{n-1} \left\| \boldsymbol{E_x}\boldsymbol{x}(t_{i+1}) - \widetilde{\boldsymbol{G}}\boldsymbol{\omega}(t_i) \right\|^2$ for a fixed projection matrix $\boldsymbol{E_x} = \boldsymbol{E}_{\mathrm{DMDc}} = \widehat{\boldsymbol{U}}_{\boldsymbol{Y}}^\top$. Here, $\boldsymbol{\omega}(t_i)$ is the concatenated vector of state and actuation as defined in section 3.3. The optimal solution $\widetilde{\boldsymbol{G}}_{\mathrm{opt}}$ is then partitioned as $[\widetilde{\boldsymbol{A}} \ \ \widetilde{\boldsymbol{B}}]$ such that $\widetilde{\boldsymbol{A}} \in \mathbb{R}^{r_x \times d_x}, \widetilde{\boldsymbol{B}} \in \mathbb{R}^{r_x \times d_u}$. Finally, $\widetilde{\boldsymbol{A}}$ is post-multiplied with the reconstruction operator $\boldsymbol{D}_{\mathrm{DMDc}} = \widehat{\boldsymbol{U}}_{\boldsymbol{Y}}$ to get the ROM components $\boldsymbol{A}_{\mathrm{R,DMDc}}$ and $\boldsymbol{B}_{\mathrm{R,DMDc}}$. Details of this process along with the proofs are given in appendix A.5. Note, the final step of this process offers dimensionality reduction only for the linear case, not in the case when the projection and reconstruction operators are nonlinear (e.g. DNNs). Therefore, we use an alternative formulation with the following results to design a loss function that encourages a DMDc-like solution for (11) and also offers dimensionality reduction when nonlinear components are used.

**Theorem 4.1.1.** *Consider the following objective function*

$$L_{\mathrm{pred}}(\boldsymbol{E_x}, \boldsymbol{G}) = \frac{1}{n} \sum_{i=0}^{n-1} \left\| \boldsymbol{E_x}\boldsymbol{x}(t_{i+1}) - \boldsymbol{G}\boldsymbol{E_{xu}}\boldsymbol{\omega}(t_i) \right\|^2, \quad (12)$$

*where* $\boldsymbol{G} = [\boldsymbol{A}_{\mathrm{R}} \ \ \boldsymbol{B}_{\mathrm{R}}] \in \mathbb{R}^{r_x \times (r_x + d_u)}, \boldsymbol{E_{xu}} = \begin{bmatrix} \boldsymbol{E_x} & \boldsymbol{0} \\ \boldsymbol{0} & \boldsymbol{I}_{d_u} \end{bmatrix} \in \mathbb{R}^{(r_x + d_u) \times (d_x + d_u)}, \boldsymbol{I}_{d_u}$ *being the identity matrix of order* $d_u$. *For any fixed matrix* $\boldsymbol{E_x}$, *the objective function* $L_{\mathrm{pred}}$ *is convex in the coefficients of* $\boldsymbol{G}$ *and attains its minimum for any* $\boldsymbol{G}$ *satisfying*

$$\boldsymbol{G}\boldsymbol{E_{xu}}\boldsymbol{\Omega}\boldsymbol{\Omega}^\top\boldsymbol{E_{xu}}^\top = \boldsymbol{E_x}\boldsymbol{Y}\boldsymbol{\Omega}^\top\boldsymbol{E_{xu}}^\top, \quad (13)$$

*where* $\boldsymbol{Y}$ *and* $\boldsymbol{\Omega}$ *are the data matrices as defined in section (3.3). If* $\boldsymbol{E_x}$ *has full rank* $r_x$, *and* $\boldsymbol{\Omega}\boldsymbol{\Omega}^\top$ *is non-singular, then* $L_{\mathrm{pred}}$ *is strictly convex and has a unique minimum for*

$$\boldsymbol{G} = [\boldsymbol{A}_{\mathrm{R}} \ \ \boldsymbol{B}_{\mathrm{R}}] = \boldsymbol{E_x}\boldsymbol{Y}\boldsymbol{\Omega}^\top\boldsymbol{E_{xu}}^\top(\boldsymbol{E_{xu}}\boldsymbol{\Omega}\boldsymbol{\Omega}^\top\boldsymbol{E_{xu}}^\top)^{-1}. \quad (14)$$

*Proof sketch.* This can be proved by a method similar to the one used for deriving the solution of linear autoencoder in (Baldi & Hornik (1989)). For any fixed $\boldsymbol{E_x}$, the objective function of (12) can be written as $L_{\mathrm{pred}}(\boldsymbol{E_x}, \boldsymbol{G}) = \left\| \mathrm{vec}(\boldsymbol{E_x}\boldsymbol{Y}) - (\boldsymbol{\Omega}^\top\boldsymbol{E_{xu}}^\top \otimes \boldsymbol{I}_{r_x})\mathrm{vec}(\boldsymbol{G}) \right\|^2$, where $\otimes$ denotes the Kronecker product and $\mathrm{vec}(\cdot)$ denotes vectorization of a matrix. Optimizing this linear least-square problem, we get (13) and (14), given the stated conditions are satisfied. The complete proof is given in appendix A.1.

**Remark.** For a unique solution, we assume that $\boldsymbol{E_x}$ has full rank. The other scenario, i.e., $\boldsymbol{E_x}$ is rank-deficient suggests poor utilization of the hidden units of the model. In that case, the number of hidden units (which represents the dimension of the reduced state) should be decreased. The assumption that the covariance matrix $\boldsymbol{\Omega\Omega}^\top$ is invertible can be ensured when $n \geq d_{\boldsymbol{x}} + d_{\boldsymbol{u}}$, by removing any linearly dependent features in system state and actuation. When $n < d_{\boldsymbol{x}} + d_{\boldsymbol{u}}$, the covariance matrix $\boldsymbol{\Omega\Omega}^\top$ is not invertible. However, similar results can be obtained by adding $\ell_2$ regularization (for the coefficients/entries of $\boldsymbol{G}$) to the objective function. Proof of this is given in appendix A.4.

### 4.1.2 The connection between the solutions of the linear autoencoding model and DMDc

The connection between the ROM obtained by minimizing $L_{\text{pred}}$ (for a fixed $\boldsymbol{E_x}$), i.e., (14) and the DMDc ROM of (10b) is not readily apparent. To interpret the connection, we formulate an alternative representation of (14) utilizing the SVD and the Moore-Penrose inverse of matrices. This alternative representation leads to the following result.

**Corollary 4.1.1.1.** *Consider the (full) SVD of the data matrix $\boldsymbol{\Omega}$ given by $\boldsymbol{\Omega} = \boldsymbol{U_\Omega \Sigma_\Omega V_\Omega}^\top$, where $\boldsymbol{U_\Omega} \in \mathbb{R}^{(d_{\boldsymbol{x}}+d_{\boldsymbol{u}}) \times (d_{\boldsymbol{x}}+d_{\boldsymbol{u}})}, \boldsymbol{\Sigma_\Omega} \in \mathbb{R}^{(d_{\boldsymbol{x}}+d_{\boldsymbol{u}}) \times n}$, and $\boldsymbol{V_\Omega} \in \mathbb{R}^{n \times n}$. If $\boldsymbol{E_x} = \widehat{\boldsymbol{U}}_{\boldsymbol{Y}}^\top$ and $\boldsymbol{\Omega\Omega}^\top$ is non-singular, then the solution for $\boldsymbol{G} = [\boldsymbol{A}_{\text{R}} \quad \boldsymbol{B}_{\text{R}}]$ corresponding to the unique minimum of $L_{pred}$ can be expressed as*

$$\boldsymbol{A}_R = \widehat{\boldsymbol{U}}_{\boldsymbol{Y}}^\top \boldsymbol{Y} \boldsymbol{V_\Omega} \Sigma^* \boldsymbol{U}_{\boldsymbol{\Omega},1}^\top \widehat{\boldsymbol{U}}_{\boldsymbol{Y}}, \quad and \quad \boldsymbol{B}_R = \widehat{\boldsymbol{U}}_{\boldsymbol{Y}}^\top \boldsymbol{Y} \boldsymbol{V_\Omega} \Sigma^* \boldsymbol{U}_{\boldsymbol{\Omega},2}^\top, \tag{15}$$

*where $[\boldsymbol{U}_{\boldsymbol{\Omega},1}^\top \quad \boldsymbol{U}_{\boldsymbol{\Omega},2}^\top] = \boldsymbol{U_\Omega}^\top$ with $\boldsymbol{U}_{\boldsymbol{\Omega},1} \in \mathbb{R}^{d_{\boldsymbol{x}} \times (d_{\boldsymbol{x}}+d_{\boldsymbol{u}})}, \boldsymbol{U}_{\boldsymbol{\Omega},2} \in \mathbb{R}^{d_{\boldsymbol{u}} \times (d_{\boldsymbol{x}}+d_{\boldsymbol{u}})}$, and $\Sigma^* = \lim_{\varepsilon \to 0} (\boldsymbol{\Sigma_\Omega}^\top \boldsymbol{U}_{\boldsymbol{\Omega},1}^\top \widehat{\boldsymbol{U}}_{\boldsymbol{Y}} \widehat{\boldsymbol{U}}_{\boldsymbol{Y}}^\top \boldsymbol{U}_{\boldsymbol{\Omega},1} \boldsymbol{\Sigma_\Omega} + \boldsymbol{\Sigma_\Omega}^\top \boldsymbol{U}_{\boldsymbol{\Omega},2}^\top \boldsymbol{U}_{\boldsymbol{\Omega},2} \boldsymbol{\Sigma_\Omega} + \varepsilon^2 \boldsymbol{I}_n)^{-1} \boldsymbol{\Sigma_\Omega}^\top$.*

*Proof sketch.* This can be derived by plugging $\boldsymbol{E_x} = \widehat{\boldsymbol{U}}_{\boldsymbol{Y}}^\top$ into (14), and using the *SVD definition* and the *limit definition* (Albert (1972)) of the Moore-Penrose inverse. The complete proof is given in appendix A.3 that uses some preliminary results presented in appendix A.2.

**Remark.** It can be verified easily that if we use the truncated SVD (as defined by 9), instead of the full SVD, for $\boldsymbol{\Omega}$ in corollary 4.1.1.1, we get an approximation of (15):

$$\widehat{\boldsymbol{A}}_{\text{R}} = \widehat{\boldsymbol{U}}_{\boldsymbol{Y}}^\top \boldsymbol{Y} \widehat{\boldsymbol{V}}_{\boldsymbol{\Omega}} \widehat{\Sigma}^* \widehat{\boldsymbol{U}}_{\boldsymbol{\Omega},1}^\top \widehat{\boldsymbol{U}}_{\boldsymbol{Y}}, \quad and \quad \widehat{\boldsymbol{B}}_{\text{R}} = \widehat{\boldsymbol{U}}_{\boldsymbol{Y}}^\top \boldsymbol{Y} \widehat{\boldsymbol{V}}_{\boldsymbol{\Omega}} \widehat{\Sigma}^* \widehat{\boldsymbol{U}}_{\boldsymbol{\Omega},2}^\top, \tag{16}$$

where $\widehat{\Sigma}^* = \lim_{\varepsilon \to 0} (\widehat{\boldsymbol{\Sigma}}_{\boldsymbol{\Omega}}^\top \widehat{\boldsymbol{U}}_{\boldsymbol{\Omega},1}^\top \widehat{\boldsymbol{U}}_{\boldsymbol{Y}} \widehat{\boldsymbol{U}}_{\boldsymbol{Y}}^\top \widehat{\boldsymbol{U}}_{\boldsymbol{\Omega},1} \widehat{\boldsymbol{\Sigma}}_{\boldsymbol{\Omega}} + \widehat{\boldsymbol{\Sigma}}_{\boldsymbol{\Omega}}^\top \widehat{\boldsymbol{U}}_{\boldsymbol{\Omega},2}^\top \widehat{\boldsymbol{U}}_{\boldsymbol{\Omega},2} \widehat{\boldsymbol{\Sigma}}_{\boldsymbol{\Omega}} + \varepsilon^2 \boldsymbol{I}_{r_{\boldsymbol{xu}}})^{-1} \widehat{\boldsymbol{\Sigma}}_{\boldsymbol{\Omega}}^\top$. We can see that (16) has the same form as (10b), except $\widehat{\boldsymbol{\Sigma}}_{\boldsymbol{\Omega}}^{-1}$ is replaced with $\widehat{\Sigma}^*$.

All the aforementioned results are derived for a fixed $\boldsymbol{E_x}$ and the relation to the DMDc is specific to the case $\boldsymbol{E_x} = \widehat{\boldsymbol{U}}_{\boldsymbol{Y}}^\top$. Note that the columns of the $\widehat{\boldsymbol{U}}_{\boldsymbol{Y}}$ are the left singular vectors, corresponding to the leading singular values, of $\boldsymbol{Y}$. Equivalently, those are also the eigenvectors, corresponding to the leading eigenvalues, of the covariance matrix $\boldsymbol{YY}^\top$. $L_{\text{pred}}$ alone does not constrain $\boldsymbol{E_x}$ to take a similar form and we need another loss term to encourage such form for the encoder. To this end, we follow the work of Baldi & Hornik (1989) on the similarity between principle component analysis and linear autoencoders, optimized with the following objective function:

$$L_{\text{recon}}(\boldsymbol{E_x}, \boldsymbol{D_x}) = \frac{1}{n} \sum_{i=1}^{n} \left\| \boldsymbol{x}(t_i) - \boldsymbol{D_x} \boldsymbol{E_x} \boldsymbol{x}(t_i) \right\|^2. \tag{17}$$

They showed that all the critical points of $L_{\text{recon}}$ correspond to projections onto subspaces associated with subsets of eigenvectors of the covariance matrix $\boldsymbol{YY}^\top$. Moreover, $L_{\text{recon}}$ has a unique global minimum corresponding to the first $r_{\boldsymbol{x}}$ (i.e., the desired dimension of the reduced state) number of eigenvectors of $\boldsymbol{YY}^\top$, associated with the leading $r_{\boldsymbol{x}}$ eigenvalues. In other words, for any invertible matrix $\boldsymbol{C} \in \mathbb{R}^{r_{\boldsymbol{x}} \times r_{\boldsymbol{x}}}$, $\boldsymbol{D_x} = \boldsymbol{U}_{r_{\boldsymbol{x}}} \boldsymbol{C}$ and $\boldsymbol{E_x} = \boldsymbol{C}^{-1} \boldsymbol{U}_{r_{\boldsymbol{x}}}^\top$ globally minimizes $L_{\text{recon}}$, where $\boldsymbol{U}_{r_{\boldsymbol{x}}}$ denotes the matrix containing leading $r_{\boldsymbol{x}}$ eigenvectors of $\boldsymbol{YY}^\top$. Since the left singular vectors of $\boldsymbol{Y}$ are the eigenvectors of $\boldsymbol{YY}^\top$, we have $\boldsymbol{U}_{r_{\boldsymbol{x}}} = \widehat{\boldsymbol{U}}_{\boldsymbol{Y}}$. Hence, we consider to utilize $L_{\text{recon}}$ to promote learning an encoder $\boldsymbol{E_x}$ in the form of $\boldsymbol{C}^{-1} \widehat{\boldsymbol{U}}_{\boldsymbol{Y}}^\top$.

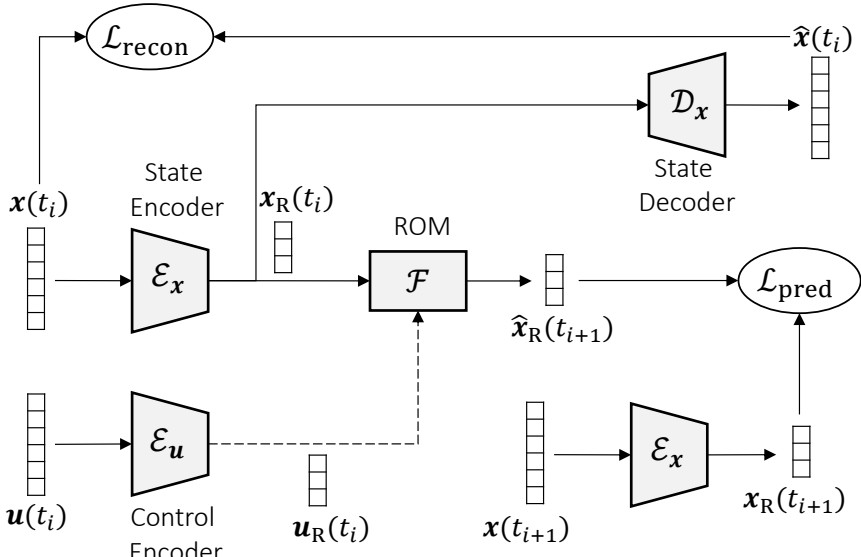

Figure 2: Autoencoding architecture for reduced order modeling. The state encoder $\mathcal{E}_{\boldsymbol{x}}$ and control encoder $\mathcal{E}_{\boldsymbol{u}}$ reduce the dimension of the state and actuation, respectively. The ROM $\mathcal{F}$ takes the current reduced state and actuation to predict the next reduced state, which is then uplifted to the full state by the state decoder $\mathcal{D}_{\boldsymbol{x}}$. All modules are trained together using a combined loss involving $\mathcal{L}_{\text{pred}}$ and $\mathcal{L}_{\text{recon}}$. The dashed arrow indicates that the $\mathcal{E}_{\boldsymbol{u}}$ is used only when $d_{\boldsymbol{u}} >> 1$; otherwise, the actuation is used as a direct input to ROM.

Accordingly, we propose to minimize the following objective function to encourage a DMDc-like solution for LAROM:

$$L(\boldsymbol{E}_{\boldsymbol{x}}, \boldsymbol{D}_{\boldsymbol{x}}, \boldsymbol{G}) = L_{\text{pred}}(\boldsymbol{E}_{\boldsymbol{x}}, \boldsymbol{G}) + \beta_1 L_{\text{recon}}(\boldsymbol{E}_{\boldsymbol{x}}, \boldsymbol{D}_{\boldsymbol{x}}), \tag{18}$$

where $\beta_1 > 0$ is a tunable hyperparameter.

It is important to note that $L_{\text{recon}}$ is minimized for any invertible matrix $\boldsymbol{C}$, $\boldsymbol{D}_{\boldsymbol{x}} = \widehat{\boldsymbol{U}}_{\boldsymbol{Y}} \boldsymbol{C}$, and $\boldsymbol{E}_{\boldsymbol{x}} = \boldsymbol{C}^{-1} \widehat{\boldsymbol{U}}_{\boldsymbol{Y}}^{\top}$. When optimized using gradient descent, it is highly unlikely to get $\boldsymbol{C}$ as the identity matrix like DMDc. Rather, we expect a random $\boldsymbol{C}$. Therefore, we need additional constraints to promote similarity with DMDc. For this purpose, we tie the matrices $\boldsymbol{E}_{\boldsymbol{x}}$ and $\boldsymbol{D}_{\boldsymbol{x}}$ to be the transpose of each other and add a semi-orthogonality constraint $\beta_4 \|\boldsymbol{E}_{\boldsymbol{x}} \boldsymbol{E}_{\boldsymbol{x}}^{\top} - \boldsymbol{I}_{r_x}\|, \beta_4 > 0$ to the optimization objective of (18).

### 4.1.3 Extending the linear model to a deep model

Here, we discuss the process of extending LAROM to a nonlinear reduced order modeling framework. We replace all the trainable components of LAROM, i.e., $\boldsymbol{E}_{\boldsymbol{x}}, \boldsymbol{D}_{\boldsymbol{x}}$, and $\boldsymbol{G}$, with DNNs. Specifically, we use an encoding function or *encoder* $\mathcal{E}_{\boldsymbol{x}} : \mathbb{X} \to \mathbb{R}^{r_x}$ and a decoding function or *decoder* $\mathcal{D}_{\boldsymbol{x}} : \mathbb{R}^{r_x} \to \mathbb{X}$ to transform the high-dimensional system state to low-dimensional features and reconstruct it back, respectively, i.e.,

$$\boldsymbol{x}_{\text{R}} = \mathcal{E}_{\boldsymbol{x}}(\boldsymbol{x}), \quad \hat{\boldsymbol{x}} = \mathcal{D}_{\boldsymbol{x}}(\boldsymbol{x}_{\text{R}}), \tag{19}$$

where $\boldsymbol{x}_{\text{R}} \in \mathbb{R}^{r_x}$ denotes the reduced state, and $\hat{\boldsymbol{x}}$ is the reconstruction of $\boldsymbol{x}$. Unlike the linear case, we use an encoder $\mathcal{E}_{\boldsymbol{u}} : \mathbb{U} \to \mathbb{R}^{r_u}, r_{\boldsymbol{u}} << d_{\boldsymbol{u}}$ for the actuation as well, in cases where the control space is also high-dimensional (for example, distributed control of spatiotemporal PDEs). The control encoder $\mathcal{E}_{\boldsymbol{u}}$ maps the high-dimensional actuation to a low-dimensional representation: $\boldsymbol{u}_{\text{R}} = \mathcal{E}_{\boldsymbol{u}}(\boldsymbol{u})$, where $\boldsymbol{u}_{\text{R}} \in \mathbb{R}^{r_u}$ denotes the encoded actuation. The encoded state and control are then fed to another DNN that represents the reduced order dynamics

$$\frac{d\boldsymbol{x}_{\text{R}}}{dt} = \mathcal{F}(\boldsymbol{x}_{\text{R}}, \boldsymbol{u}_{\text{R}}), \tag{20}$$

where $\mathcal{F} : \mathbb{R}^{r_{\boldsymbol{x}}} \times \mathbb{R}^{r_{\boldsymbol{u}}} \to \mathbb{R}^{r_{\boldsymbol{x}}}$. Given the current reduced state $\boldsymbol{x}_{\mathrm{R}}(t_i)$ and control input $\boldsymbol{u}_{\mathrm{R}}(t_i)$, the next reduced state $\boldsymbol{x}_{\mathrm{R}}(t_{i+1})$ can be computed by integrating $\mathcal{F}$ using standard numerical integrator or neural ODE (Chen et al. (2018)):

$$\boldsymbol{x}_{\mathrm{R}}(t_{i+1}) = \boldsymbol{x}_{\mathrm{R}}(t_i) + \int_{t_i}^{t_{i+1}} \mathcal{F}\big(\boldsymbol{x}_{\mathrm{R}}(t_i), \boldsymbol{u}_{\mathrm{R}}(t_i)\big)dt \triangleq \mathcal{G}\big(\boldsymbol{x}_{\mathrm{R}}(t_i), \boldsymbol{u}_{\mathrm{R}}(t_i)\big). \tag{21}$$

We can say that $\mathcal{G}$ is the nonlinear counterpart of $\boldsymbol{G}$.

Note, here the ROM is represented as a continuous-time dynamics, unlike the linear case where we used a discrete-time model. We use a discrete-time formulation for LAROM to establish its similarity with DMDc, which is formulated in discrete time. DeepROM can be formulated in a similar fashion as well. However, the specific control learning algorithm we used, which will be discussed in the next subsection, requires vector fields of the learned ROM for training. Therefore, we formulate the ROM in continuous time so that it provides the vector field $\mathcal{F}(\boldsymbol{x}_{\mathrm{R}}, \boldsymbol{u}_{\mathrm{R}})$ of the dynamics. In cases where only the prediction model is of interest and control learning is not required, a discrete-time formulation should be used for faster training of the ROM.

We train $\mathcal{E}_{\boldsymbol{x}}, \mathcal{E}_{\boldsymbol{u}}, \mathcal{D}_{\boldsymbol{x}}$, and $\mathcal{F}$ by minimizing the following loss function, analogous to (18),

$$\mathcal{L}(\mathcal{E}_{\boldsymbol{x}}, \mathcal{E}_{\boldsymbol{u}}, \mathcal{D}_{\boldsymbol{x}}, \mathcal{F}) = \mathcal{L}_{\mathrm{pred}}(\mathcal{E}_{\boldsymbol{x}}, \mathcal{E}_{\boldsymbol{u}}, \mathcal{F}) + \beta_2 \mathcal{L}_{\mathrm{recon}}(\mathcal{E}_{\boldsymbol{x}}, \mathcal{D}_{\boldsymbol{x}}), \tag{22}$$

where $\beta_2 > 0$ is a tunable hyperparameter and $\mathcal{L}_{\mathrm{pred}}, \mathcal{L}_{\mathrm{recon}}$ are defined as follows,

$$\mathcal{L}_{\mathrm{pred}}(\mathcal{E}_{\boldsymbol{x}}, \mathcal{E}_{\boldsymbol{u}}, \mathcal{F}) = \frac{1}{n} \sum_{i=0}^{n-1} \left\| \mathcal{E}_{\boldsymbol{x}}\big(\boldsymbol{x}(t_{i+1})\big) - \mathcal{G}\Big(\mathcal{E}_{\boldsymbol{x}}\big(\boldsymbol{x}(t_i)\big), \mathcal{E}_{\boldsymbol{u}}\big(\boldsymbol{u}(t_i)\big)\Big) \right\|^2,$$

$$\mathcal{L}_{\mathrm{recon}}(\mathcal{E}_{\boldsymbol{x}}, \mathcal{D}_{\boldsymbol{x}}) = \frac{1}{n} \sum_{i=1}^{n} \left\| \boldsymbol{x}(t_i) - \mathcal{D}_{\boldsymbol{x}} \circ \mathcal{E}_{\boldsymbol{x}}\big(\boldsymbol{x}(t_i)\big) \right\|^2. \tag{23}$$

Here, the operator $\circ$ denotes the composition of two functions. In experiments, $\mathcal{L}_{\mathrm{recon}}$ also includes the reconstruction loss of the desired state where we want to stabilize the system. Figure 2 shows the overall framework for training DeepROM.

## 4.2 Learning control

Once we get a trained ROM of the form (20) using the method proposed in section 4.1, the next goal is to design a controller for the system utilizing that ROM. Since our ROM is represented by DNNs, we need a data-driven method to develop the controller. We adopt the approach presented by Saha et al. (2021) for learning control law for nonlinear systems, represented by DNNs. The core idea of the method is to hypothesize a target dynamics that is exponentially stable at the desired state and simultaneously learn a control policy to realize that target dynamics in closed loop. A DNN is used to represent the vector field $\mathcal{F}_s : \mathbb{R}^{r_{\boldsymbol{x}}} \to \mathbb{R}^{r_{\boldsymbol{x}}}$ of the target dynamics $\frac{d\boldsymbol{x}_{\mathrm{R}}}{dt} = \mathcal{F}_s(\boldsymbol{x}_{\mathrm{R}})$. We use another DNN to represent a controller $\Pi : \mathbb{R}^{r_{\boldsymbol{x}}} \to \mathbb{R}^{d_{\boldsymbol{u}}}$ that provides the necessary actuation for a given reduced state $\boldsymbol{x}_{\mathrm{R}}$:

$$\boldsymbol{u} = \Pi(\boldsymbol{x}_{\mathrm{R}}). \tag{24}$$

This control $\boldsymbol{u}$ is then encoded by (trained) $\mathcal{E}_{\boldsymbol{u}}$ to its low-dimensional representation $\boldsymbol{u}_{\mathrm{R}}$. Finally, the reduced state $\boldsymbol{x}_{\mathrm{R}}$ and actuation $\boldsymbol{u}_{\mathrm{R}}$ are fed to the (trained) ROM of (20) to get $\mathcal{F}(\boldsymbol{x}_{\mathrm{R}}, \boldsymbol{u}_{\mathrm{R}})$. The overall framework for learning control is shown in Figure 3.

Our training objective is to minimize the difference between $\mathcal{F}(\boldsymbol{x}_{\mathrm{R}}, \boldsymbol{u}_{\mathrm{R}})$ and $\mathcal{F}_s(\boldsymbol{x}_{\mathrm{R}})$, i.e.,

$$\mathcal{L}_{\mathrm{ctrl}}(\mathcal{F}_s, \Pi) = \frac{1}{n} \sum_{i=1}^{n} \left\| \mathcal{F}\big(\mathcal{E}_{\boldsymbol{x}}(\boldsymbol{x}(t_i)), \mathcal{E}_{\boldsymbol{u}} \circ \Pi \circ \mathcal{E}_{\boldsymbol{x}}(\boldsymbol{x}(t_i))\big) - \mathcal{F}_s \circ \mathcal{E}_{\boldsymbol{x}}\big(\boldsymbol{x}(t_i)\big) \right\|^2. \tag{25}$$

To minimize the control effort, we add a regularization loss with (25), and the overall training objective for learning control is given by

$$\mathcal{L}_{\mathrm{ctrl,reg}}(\mathcal{F}_s, \Pi) = \mathcal{L}_{\mathrm{ctrl}}(\mathcal{F}_s, \Pi) + \beta_3 \frac{1}{n} \sum_{i=1}^{n} \left\| \Pi(\boldsymbol{x}_{\mathrm{R}}(t_i)) \right\|^2, \tag{26}$$

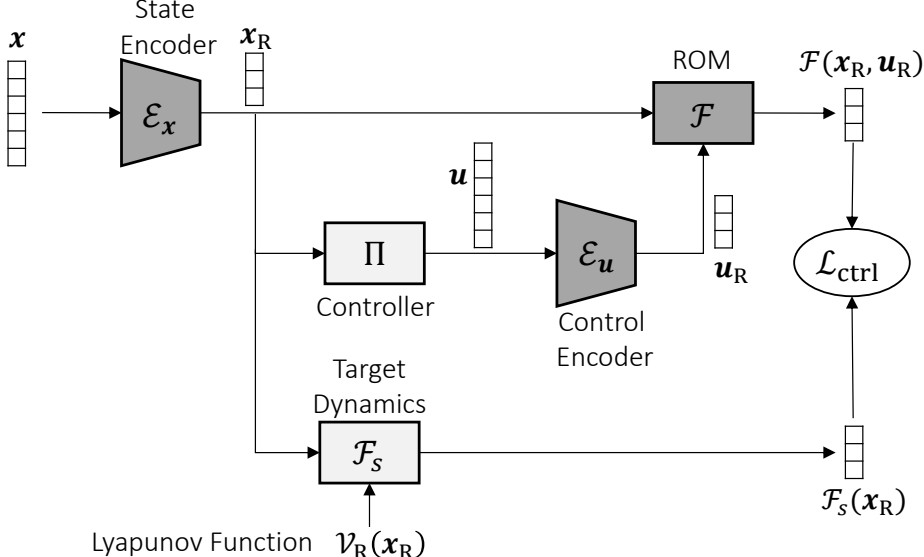

Figure 3: The control learning process. Given a reduced state, $\mathcal{F}_s$ predicts a target dynamics for the closed-loop system, and the controller $\Pi$ predicts an actuation to achieve that target. Both the modules are trained jointly using the loss function $\mathcal{L}_{\text{ctrl}}$. Parameters of the dark-shaded modules are kept fixed during this process.

where $\beta_3 > 0$ is a tunable hyperparameter. Here we jointly train the DNNs representing $\Pi$ and $\mathcal{F}_s$ only, whereas the previously-trained DNNs for $\mathcal{E}_{\boldsymbol{x}}, \mathcal{E}_{\boldsymbol{u}}$, and $\mathcal{F}$ are kept frozen. Once all the DNNs are trained, we only need $\mathcal{E}_{\boldsymbol{x}}$ and $\Pi$ during evaluation to generate actuation for the actual system, given a full-state observation:

$$\boldsymbol{u} = \Pi \circ \mathcal{E}_{\boldsymbol{x}}(\boldsymbol{x}) = \pi(\boldsymbol{x}). \tag{27}$$

As we mentioned earlier, we require the target dynamics, hypothesized by a DNN, to be exponentially stable at the desired state. Without loss of generality, we consider stability at $\boldsymbol{x}_{\text{R}} = \boldsymbol{0}$. As we mentioned earlier, the system can be stabilized at any desired state by adding a feedforward component to the control. Dynamics represented by a standard neural network is not stable at any equilibrium point, in general. Kolter & Manek (2019) showed that it is possible to design a DNN, by means of Lyapunov functions, to represent a dynamics that is exponentially stable at an equilibrium point. Accordingly, we represent our target dynamics as follows:

$$\frac{d\boldsymbol{x}_{\text{R}}}{dt} = \mathcal{F}_s(\boldsymbol{x}_{\text{R}}) = \mathcal{P}(\boldsymbol{x}_{\text{R}}) - \frac{\text{ReLU}\big(\nabla\mathcal{V}_{\text{R}}(\boldsymbol{x}_{\text{R}})^\top \mathcal{P}(\boldsymbol{x}_{\text{R}}) + \alpha\mathcal{V}_{\text{R}}(\boldsymbol{x}_{\text{R}})\big)}{\|\nabla\mathcal{V}_{\text{R}}(\boldsymbol{x}_{\text{R}})\|^2} \nabla\mathcal{V}_{\text{R}}(\boldsymbol{x}_{\text{R}}), \tag{28}$$

where $\alpha$ is a positive constant, $\text{ReLU}(z) = \max(0, z),\ z \in \mathbb{R}$, and $\mathcal{V}_{\text{R}} : \mathbb{R}^{r_{\boldsymbol{x}}} \to \mathbb{R}$ is a candidate Lyapunov function, i.e., satisfies the criteria similar to (3) and (5). We use

$$\mathcal{V}_{\text{R}}(\boldsymbol{x}_{\text{R}}) = \boldsymbol{x}_{\text{R}}^\top \boldsymbol{K} \boldsymbol{x}_{\text{R}}, \tag{29}$$

where $\boldsymbol{K} \in \mathbb{R}^{r_{\boldsymbol{x}} \times r_{\boldsymbol{x}}}$ is a positive definite matrix.

Though the efficacy of learning control by minimizing the difference with respect to a target dynamics is experimentally demonstrated by Saha et al. (2021), the stability of the closed-loop system subjected to the learned control law has not been studied analytically. Here, we present a result that shows that if we can minimize $\mathcal{L}_{\text{ctrl}}$ such that the difference between the target dynamics and the closed-loop dynamics is sufficiently small for all $\boldsymbol{x}_{\text{R}} \in \mathbb{X}_{\text{R}} \subset \mathbb{R}^{r_{\boldsymbol{x}}}$, then the trajectories of the closed-loop ROM starting sufficiently close to the origin remains close to the origin, i.e., *ultimately bounded* (Khalil (2002)). Boundedness of the closed-loop ROM trajectories under the proposed control policy is a necessary but not sufficient requirement for the stability of the original system.

**Theorem 4.2.1.** *Consider the target dynamics defined by (28) and the candidate Lyapunov function defined by (29). Suppose the difference between the target dynamics and the closed-loop dynamics satisfies*

$$\|\mathcal{F}(\boldsymbol{x}_{\mathrm{R}}, \mathcal{E}_{\boldsymbol{u}} \circ \Pi(\boldsymbol{x}_{\mathrm{R}})) - \mathcal{F}_s(\boldsymbol{x}_{\mathrm{R}})\| \leq \delta < \frac{\alpha\theta\lambda_{\min}(\boldsymbol{K})}{2\lambda_{\max}(\boldsymbol{K})}\sqrt{\frac{\lambda_{\min}(\boldsymbol{K})}{\lambda_{\max}(\boldsymbol{K})}}\eta, \tag{30}$$

*for all $\boldsymbol{x}_{\mathrm{R}} \in \mathbb{X}_{\mathrm{R}} = \{\boldsymbol{x}_{\mathrm{R}} \in \mathbb{R}^{r_{\boldsymbol{x}}} \mid \|\boldsymbol{x}_{\mathrm{R}}\| < \eta\}$ and $0 < \theta < 1$. Then, for all initial points satisfying $\|\boldsymbol{x}_{\mathrm{R}}(t_0)\| < \sqrt{\frac{\lambda_{\min}(\boldsymbol{K})}{\lambda_{\max}(\boldsymbol{K})}}\eta$, the solution of the closed-loop ROM $\frac{d\boldsymbol{x}_{\mathrm{R}}}{dt} = \mathcal{F}(\boldsymbol{x}_{\mathrm{R}}, \mathcal{E}_{\boldsymbol{u}} \circ \Pi(\boldsymbol{x}_{\mathrm{R}}))$ satisfies*

$$\|\boldsymbol{x}_{\mathrm{R}}(t)\| \leq \lambda e^{-\gamma(t-t_0)}\|\boldsymbol{x}_{\mathrm{R}}(t_0)\|, \quad \forall\, t_0 \leq t < t_c + t_0 \tag{31}$$

*and*

$$\|\boldsymbol{x}_{\mathrm{R}}(t)\| \leq \frac{2\delta}{\alpha\theta}\lambda^3, \quad \forall\, t \geq t_c + t_0 \tag{32}$$

*for some finite $t_c > 0$, where*

$$\gamma = \frac{\alpha(1-\theta)\lambda_{\min}(\boldsymbol{K})}{2\lambda_{\max}(\boldsymbol{K})} \quad and \quad \lambda = \sqrt{\frac{\lambda_{\max}(\boldsymbol{K})}{\lambda_{\min}(\boldsymbol{K})}} \tag{33}$$

*Proof Sketch.* This can be proved by first deriving the Lyapunov conditions for the target dynamics (28) (Theorem 1, Kolter & Manek (2019)) and then applying the stability analysis of perturbed systems (Lemma 9.2, Khalil (2002)) and ultimate boundedness (Theorem 4.18, Khalil (2002)) on the closed-loop ROM. A unified proof is provided in appendix A.6.

## 5 Empirical Results

For empirical analysis, we consider modeling and controlling spatiotemporal PDE-driven systems with high-dimensional measurements over discretized space. One of the primary applications of reduced order modeling lies in comprehending the behavior of complex physical processes which are typically characterized by systems of PDEs. Since spatiotemporal PDE-driven systems are infinite-dimensional in their continuous form and high-dimensional when discretized, they are a fitting choice for evaluating our method. The first example investigates a single variable actuation, whereas distributed actuation is considered for the second example.

### 5.1 Baselines

The similarity between DMDc and LAROM is demonstrated using the dynamic modes estimated in respective methods. The prediction performance of DeepROM is compared against DMDc and the Deep Koopman model (Morton et al. (2018)). The Deep Koopman model shares a similar DNN-based autoencoding structure as ours, with the distinction that its (reduced order) dynamic model is linear. The method proposed by Morton et al. (2018) considers a model predictive scenario, where the state/system matrix of the linear reduced order model is updated with online observations during operation while the input/control matrix is kept fixed. However, in contrast to the original method, we keep both matrices fixed during operation (once those are trained) as we consider offline control design in this paper. For the same reason, we apply *linear quadratic regulator* (LQR) on the ROM obtained from the Deep Koopman method, instead of model predictive control, to compare the control performance with our method: DeepROC. The control performance is also compared against the reduced order controller obtained by applying LQR on the ROM derived from DMDc.

Details on the neural network architectures and training settings for the Deep Koopman model are given in appendix D.

## 5.2 Reaction–diffusion system stabilization

For the first experiment, we consider the Newell–Whitehead–Segel reaction-diffusion equation with the Neumann boundary condition

$$\frac{\partial q}{\partial t} = \sigma \nabla^2 q + q(1 - q^2) + \mathbf{1}_{\mathbb{W}} w \quad \text{in } \mathbb{I} \times \mathbb{R}^+,$$
$$\nabla q(\zeta_l, t) = \nabla q(\zeta_r, t) = 0, \quad t \in \mathbb{R}^+, \tag{34}$$

which is used to describe various nonlinear physical systems including Rayleigh–Bénard convection. This example is used by Kalise & Kunisch (2018) to evaluate nonlinear controllers designed from reduced order state space representation. Similar systems are used for modeling problems as well in 1D (Raissi et al. (2019)) and 2D (Li et al. (2020)). In (34), $q(\zeta, t) \in \mathbb{R}$ denotes the measurement variable such as concentration or temperature at location $\zeta \in \mathbb{I} \subset \mathbb{R}$ and time $t$; $\sigma$ denotes the diffusion coefficient; $w(t) \in \mathbb{R}$ is the actuation at time $t$ and $\mathbf{1}_{\mathbb{W}}(\zeta)$ is the indicator function with $\mathbb{W} \subset \mathbb{I}$; $\zeta_l$ and $\zeta_r$ denote the boundary points of $\mathbb{I}$. (34) is a bistable system with $\pm 1$ as stable and 0 as unstable equilibria. For the control task, we consider feedback stabilization of (34) at the unstable equilibrium 0, as studied by Kalise & Kunisch (2018). We use $\mathbb{I} = (-1, 1), \mathbb{W} = (-0.2, 0.2)$, and $\sigma = 0.2$. Details on dataset generation, neural network architectures, and training settings are given in appendix B.

### 5.2.1 Similarity with DMDc

To investigate the similarity DMDc, we first train the LAROM using gradient descent to minimize the objective (18) with the semi-orthogonality regularization and enforcing $\boldsymbol{D_x} = \boldsymbol{E_x}^\top$, as discussed in 4.1.2.

The dynamic modes for LAROM are computed as $\boldsymbol{\varphi}_i = \boldsymbol{D_x} \boldsymbol{z}_i$, where $\boldsymbol{z}_i$ is the $i^{\text{th}}$ eigenvector of $\boldsymbol{A}_{\text{R}}$. Similarly, the dynamic modes for DMDc are computed as $\boldsymbol{\varphi}_{i,\text{DMDc}} = \boldsymbol{D}_{\text{DMDc}} \boldsymbol{z}_{i,\text{DMDc}}$, where $\boldsymbol{z}_{i,\text{DMDc}}$ is the $i^{\text{th}}$ eigenvector of $\boldsymbol{A}_{\text{R,DMDc}}$. Note, these dynamic modes are similar to the ones used in the original DMD algorithm Schmid (2010), not the exact modes obtained in Proctor et al. (2016). Exact modes cannot be computed for LAROM since it does not involve SVD. Modes defined by $\boldsymbol{\varphi}_{i,\text{DMDc}} = \boldsymbol{D}_{\text{DMDc}} \boldsymbol{z}_{i,\text{DMDc}} = \widehat{\boldsymbol{U}_Y} \boldsymbol{z}_{i,\text{DMDc}}$ are the orthogonal projection of the exact modes onto the range of $\boldsymbol{Y}$ (Theorem 3, Tu et al. (2014)). Figure 4 compares the dynamic modes obtained using DMDc and LAROM for the case when the dimension of the ROMs is 3. It is important to note that the numbering of the modes is arbitrary as the optimal ranking of DMDc modes is not trivial. The correspondence between the DMDc modes and LAROM modes are determined by comparing the eigenvalues of $\boldsymbol{A}_{\text{R,DMDc}}$ and $\boldsymbol{A}_{\text{R}}$. Dynamic modes of both methods are similar except for the different signs of the first two modes.

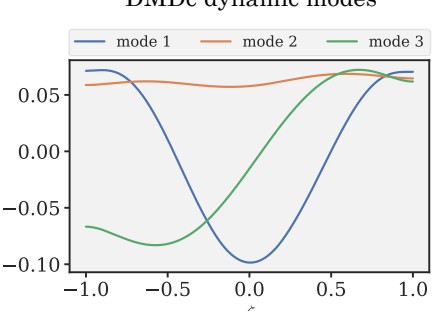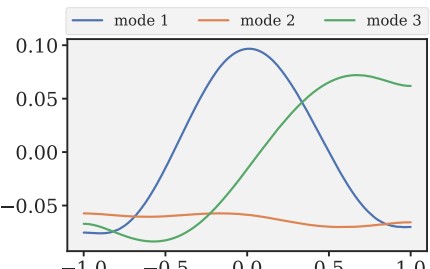

Figure 4: The first three dynamic modes of the reaction–diffusion system, obtained using DMDc and LAROM.

### 5.2.2 Prediction performance of DeepROM

We now compare the performance of DeepROM, Deep Koopman model, and DMDc in the prediction task. Note, this example uses low-dimensional actuation (just a single variable). Accordingly, the control encoder

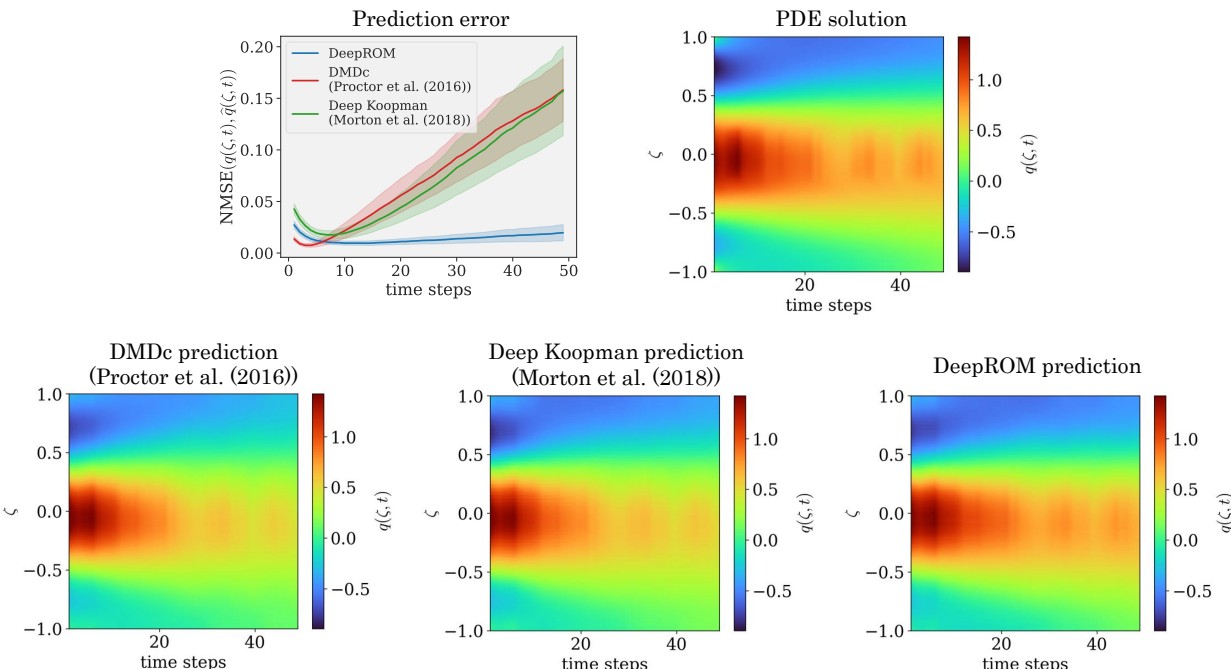

Figure 5: Prediction performance of DMDc, Deep Koopman, and DeepROM in the reaction–diffusion example. The prediction error plot shows the mean error and 95% confidence interval from 100 test sequences and for Deep Koopman and DeepROM, 3 different training instances. One example sequence is used to visually compare the predictions with the solution from a PDE solver.

$\mathcal{E}_{\boldsymbol{u}}$ is not used here. Figure 5 shows the quantitative and qualitative comparison of the recursive multi-step predictions obtained using DMDc, Deep Koopman model, and DeepROM. The prediction error is computed as *normalized mean squared error* (NMSE) with respect to the solution obtained using the PDE solver. The prediction error plot shows the mean error and 95% confidence interval from 100 test sequences and for Deep Koopman and DeepROM, 3 different training instances. The color maps are shown for one example sequence with one training instance. Prediction error increases more quickly for DMDc and Deep Koopman than DeepROM as the linear ROMs become less accurate in the long term.

### 5.2.3 Control performance of DeepROC

Figure 6 shows the control performance of DeepROC, Deep Koopman + LQR, and DMDc + LQR in the task of stabilizing the system at the unstable equilibrium 0 from an initial state $2 + \cos(2\pi\zeta)\cos(\pi\zeta)$. We use the following metrics for comparison:

(i) mean squared error over time between the controlled solutions and the desired profile

(ii) differential magnitude that measures the differential changes between the profiles at consecutive time steps. In the steady state, the differential magnitude should be close to zero.

(iii) the amount of actuation applied

For Deep Koopman and DeepROC, the plots show the mean values with 1-standard deviation interval from 3 training instances. All methods show similar closed-loop error profiles. However, DeepROC requires significantly less amount of actuation in comparison with DMDc + LQR and Deep Koopman + LQR to reach a similar steady-state error. DeepROC can account for the decaying nonlinear term $-q^3$ present in the system (34) and therefore learns to apply less actuation. Figure 7 visually compares the uncontrolled solution and the controlled solutions obtained using the three methods. When uncontrolled, the system reaches the stable equilibrium at 1, whereas the feedback-controlled system is stabilized at the desired state 0 in both cases.

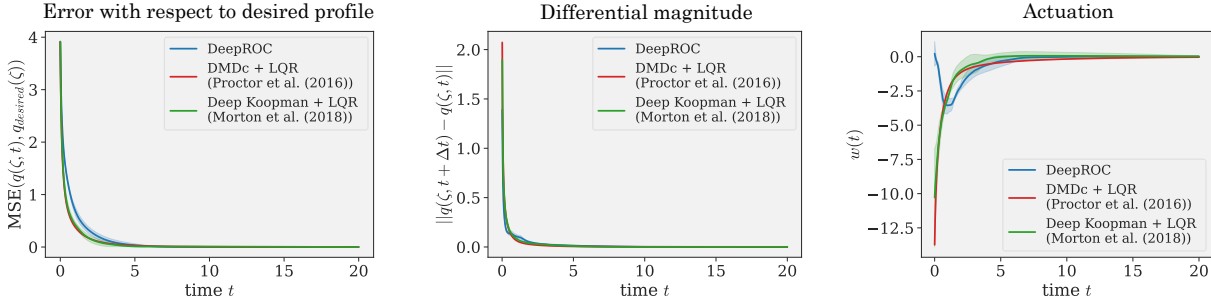

Figure 6: Control performance of DMDc + LQR, Deep Koopman + LQR, and DeepROC in the reaction–diffusion example. For Deep Koopman and DeepROC, the plots show the mean values with 1-standard deviation interval from 3 training instances.

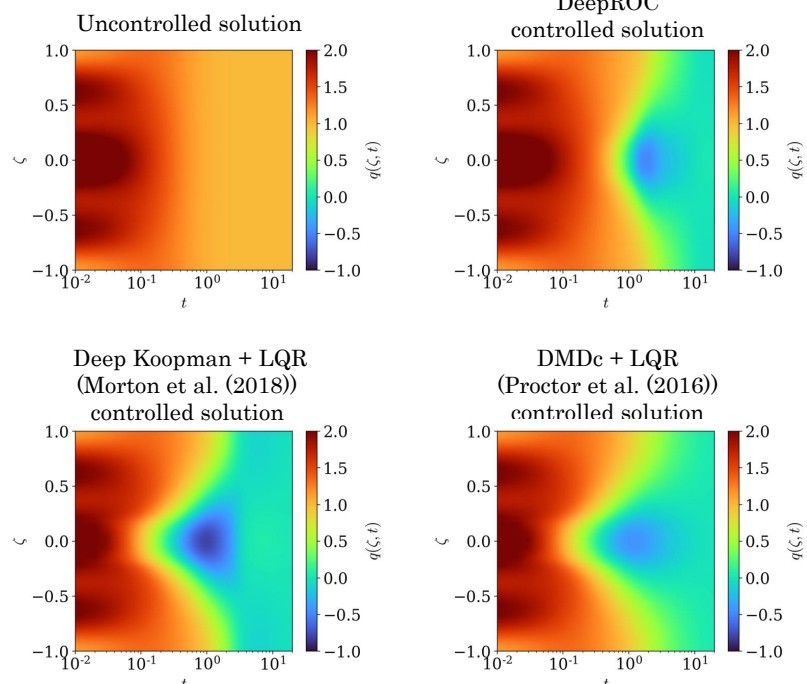

Figure 7: Visual comparison of the uncontrolled solution and the controlled solutions using DeepROC, Deep Koopman + LQR, and DMDc + LQR.

## 5.3 Vortex shedding suppression in fluid

In this experiment, we consider modeling and suppressing vortex shedding in two-dimensional incompressible flow past a circular cylinder. This is a well-known problem (Schäfer et al. (1996)) and is of great importance for many engineering applications (Williamson (1996)). Several previous studies on deep learning-based modeling and control have used this system for evaluation (Eivazi et al. (2020); Erichson et al. (2019); Rabault et al. (2019); Tang et al. (2020); Bieker et al. (2020); Morton et al. (2018)). The dynamics is governed by the incompressible Navier-Stokes equations given by

$$\frac{\partial \boldsymbol{v}}{\partial t} - \nu \nabla^2 \boldsymbol{v} + (\boldsymbol{v} \cdot \nabla)\boldsymbol{v} = -\frac{1}{\rho}\nabla p + \mathbf{1}_{\mathbb{W}}\boldsymbol{w}, \quad \nabla \cdot \boldsymbol{v} = \mathbf{0} \quad \text{in } \mathbb{I} \times \mathbb{R}^+, \tag{35}$$

where $\boldsymbol{v}(\boldsymbol{\zeta}, t) \in \mathbb{R}^2$ denotes the flow velocity at location $\boldsymbol{\zeta} \in \mathbb{I} \subset \mathbb{R}^2$ and time $t$, $p(\boldsymbol{\zeta}, t) \in \mathbb{R}$ denotes the pressure, $\nu$ denotes the kinematic viscosity and $\rho$ denotes the density of the fluid. $\boldsymbol{w}(\boldsymbol{\zeta}, t)$ is the actuation/force applied to the system and $\boldsymbol{1}_{\mathbb{W}}(\boldsymbol{\zeta})$ is the indicator function with $\mathbb{W} \subset \mathbb{I}$. We use $\mathbb{I} = (0, 2.2) \times (0, 0.41)$ and $\mathbb{W} = (0.11, 0.77) \times (0, 0.41)$. Density and kinematic viscosity are chosen such that the Reynolds number is $Re = 50$, which is just above the cutoff for the onset of the vortex shedding (Williamson (1996)). In this case, vortices are created at the back of the cylinder and are shed periodically from the upper and lower surfaces of the cylinder forming a von Kármán vortex street (Morton et al. (2018)). We use the domain $\mathbb{W}$ for observation and distributed actuation. The Stokes flow is used as the desired state for the control task. More details on the problem setup, dataset generation, neural network architectures, and training settings are given in appendix C.

### 5.3.1 Similarity with DMDc

To analyze the dynamic modes, we train the LAROM by enforcing $\boldsymbol{D_x} = \boldsymbol{E_x^\top}$ and adding the semi-orthogonality constraint to the learning objective, as mentioned in 4.1.2. Figure 8 compares the first two oscillatory dynamic modes obtained using DMDc and LAROM. Only the streamwise components are shown for brevity. Also, complex modes occur in conjugate pairs and only one from each pair is shown. The correspondence between the DMDc modes and LAROM modes are determined by comparing the eigenvalues of $\boldsymbol{A}_{\mathrm{R,DMDc}}$ and $\boldsymbol{A}_{\mathrm{R}}$. Dynamic modes identified by LAROM are similar to the ones obtained from DMDc, except the real and imaginary components of the first mode are swapped.

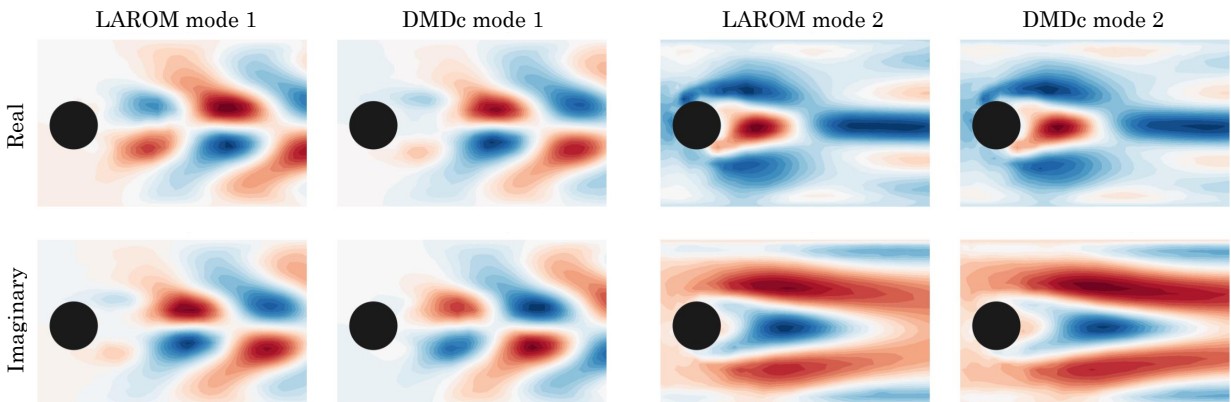

Figure 8: The first two dynamic modes obtained using DMDc and LAROM for the flow past a cylinder system.

### 5.3.2 Prediction performance of DeepROM

Figure 9 shows the quantitative and qualitative comparison of the recursive multi-step predictions, starting from $t = 0.1$, obtained using DMDc, Deep Koopman model, and DeepROM. The initial state is chosen at $t = 0.1$ because the fluid does not reach the observation region $\mathbb{W}$ before that time. The prediction error is computed as the *mean squared error* (MSE) with respect to the solution obtained using a PDE solver. For Deep Koopman and DeepROM, the prediction error plot shows the mean error and 1-standard deviation interval from 3 training instances. DeepROM shows lower prediction error in comparison with DMDc. The Deep Koopman model shows better prediction performance than DeepROM and DMDc during the initial few steps. However, its accuracy deteriorates rapidly and eventually becomes comparable to that of DMDc. Moreover, unlike DeepROM, DMDc and Deep Koopman model are unable to capture the shedding pattern in multi-step prediction as shown in the contour plots of the velocity magnitude.

### 5.3.3 Control performance of DeepROC

Figure 10 shows the control performance of DeepROC, Deep Koopman + LQR, and DMDc+LQR in the task of suppressing vortex shedding. The controllers of DeepROC and DMDc + LQR directly estimate the high-

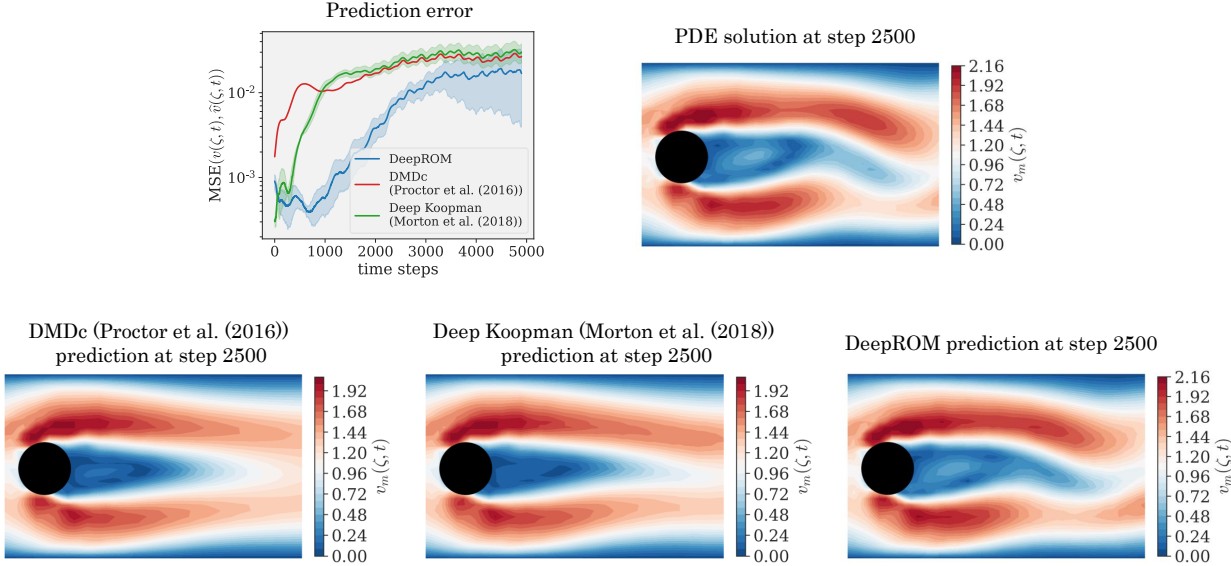

Figure 9: Prediction performance of DMDc, Deep Koopman, and DeepROM in the fluid flow example. For Deep Koopman and DeepROM, the prediction error plot shows the mean error and 1-standard deviation interval from 3 training instances. Predictions at time step 2500 for the test sequence are visually compared with the solution from a PDE solver. $v_m$ denotes the velocity magnitude.

dimensional actuation distributed over space. However, the same technique proved ineffective in suppressing the shedding for Deep Koopman + LQR. Therefore, instead of directly estimating the distributed actuation, we utilize a low-dimensional representation of the actuation for Deep Koopman + LQR. We represent the distributed actuation as a linear combination of some space-dependent sinusoidal basis functions. The controller is designed to estimate the coefficients of those basis functions in the linear combination. Details are provided in appendix D.

We use the same metrics as the previous example for comparison except for actuation. Since distributed control is applied in this case, we use the magnitude of the actuation here. For DeepROC and Deep Koopman + LQR, the plots show the mean values with 1-standard deviation interval from 3 training instances. To reach a similar steady-state error, DeepROC takes a longer time compared to DMDc and Deep Koopman + LQR. DeepROM uses the least amount of actuation during the initial few steps, whereas Deep Koopman + LQR has the least steady-state actuation magnitude. Figure 11 shows the velocity magnitude of the controlled flow for DeepROC, Deep Koopman + LQR, and DMDc+LQR at different times, starting from a von Kármán vortex street pattern. All methods accomplish a similar steady-state flow pattern where vortex shedding has been suppressed.

## 6 Conclusion

We presented a framework for autoencoder-based modeling and control learning for high-dimensional dynamical systems. We showed that autoencoding ROMs are capable of capturing the dominant modes that are essential in analyzing and designing control for the underlying systems. As we showed in experiments, DeepROM offers better prediction accuracy than a linear ROM over a relatively longer prediction horizon when applied to nonlinear systems. However, this advantage does not always translate to significant improvement in control performance. Though the used control learning method theoretically ensures ultimate boundedness for the closed-loop ROM solution, data-driven optimization of the learning objective often makes the models susceptible to distribution shift which can impact the control performance. The control learning process in the DeepROC framework can easily be replaced with other methods like model-based RL or model predictive control. It would be interesting for future work to investigate whether updating both the reduced model and the controller in the MPC framework ensures robustness under distribution shift and

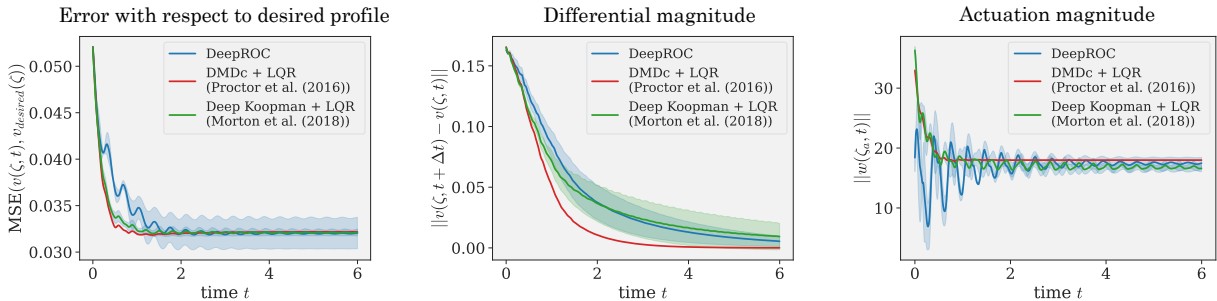

Figure 10: Control performance of DMDc + LQR, Deep Koopman + LQR, and DeepROC in the vortex shedding suppression task. For Deep Koopman + LQR and DeepROC, the plots show the mean values with 1-standard deviation interval from 3 training instances.

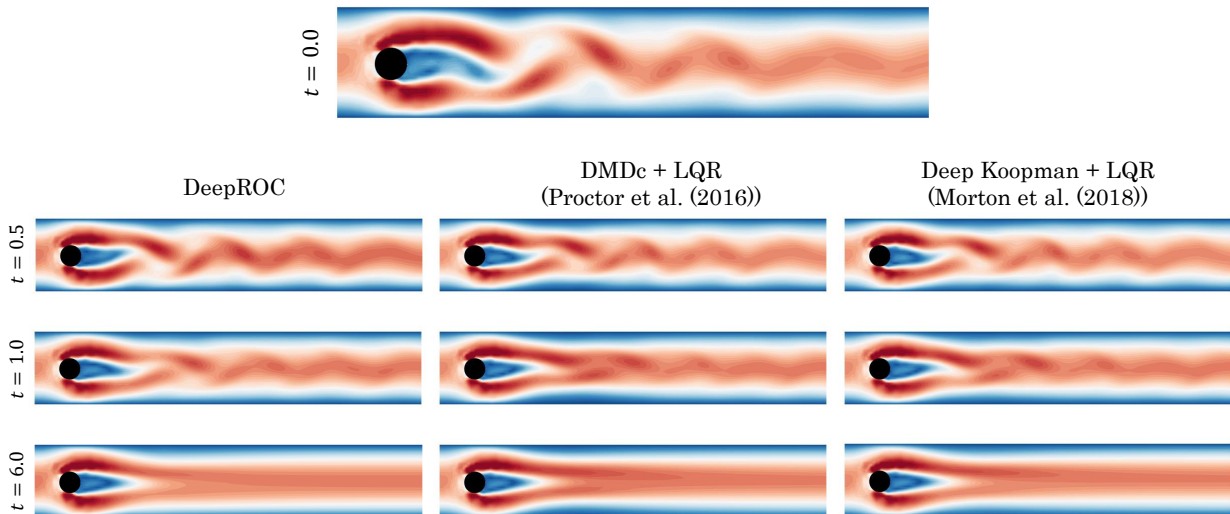

Figure 11: Visual comparison of the velocity magnitude of the flow over time subjected to the controllers obtained using DeepROC, Deep Koopman + LQR, and DMDc + LQR.

offers better control performance. Designing controllers for DNN-based models is a challenging task due to the standard difficulties associated with non-convex optimization. Nevertheless, we envision great prospects in solving many problems of control design for high-dimensional systems utilizing autoencoder-based models as they continue to demonstrate their effectiveness in the analysis and prediction of such systems.

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

# Appendices

## A   Proofs

This section details the proofs for the results presented in section 4. The proof of theorem 4.1.1 uses the following properties of the rank (denoted by $\text{rank}(\cdot)$), the Kronecker product (denoted by $\otimes$) and vectorization of matrices (denoted by $\text{vec}(\cdot)$). All the definitions and properties are presented in the context of matrices over real numbers.

For any conformable matrices $\boldsymbol{D}$ and $\boldsymbol{E}$ such that $\boldsymbol{E}$ has full row-rank,

$$\text{rank}(\boldsymbol{D}\boldsymbol{E}) = \text{rank}(\boldsymbol{D}). \tag{36a}$$

For any real matrix $\boldsymbol{D}$,

$$\text{rank}(\boldsymbol{D}^\top \boldsymbol{D}) = \text{rank}(\boldsymbol{D}\boldsymbol{D}^\top) = \text{rank}(\boldsymbol{D}^\top) = \text{rank}(\boldsymbol{D}). \tag{36b}$$

For any matrices (of compatible dimensions) $\boldsymbol{D}, \boldsymbol{E}, \boldsymbol{F}$, and $\boldsymbol{H}$,

$$\text{vec}(\boldsymbol{D}\boldsymbol{E}\boldsymbol{F}^\top) = (\boldsymbol{F} \otimes \boldsymbol{D})\text{vec}(\boldsymbol{E}), \tag{37a}$$

$$(\boldsymbol{D} \otimes \boldsymbol{E})^\top = \boldsymbol{D}^\top \otimes \boldsymbol{E}^\top, \tag{37b}$$

$$(\boldsymbol{D} \otimes \boldsymbol{E})(\boldsymbol{F} \otimes \boldsymbol{H}) = (\boldsymbol{D}\boldsymbol{F} \otimes \boldsymbol{E}\boldsymbol{H}), \tag{37c}$$

whenever these quantities are defined. Furthermore, if $\boldsymbol{D}$ and $\boldsymbol{E}$ are symmetric and positive semidefinite (resp. positive definite), then $\boldsymbol{D} \otimes \boldsymbol{E}$ is symmetric and positive semidefinite (resp. positive definite), i.e.,

$$\boldsymbol{D} \succeq 0, \boldsymbol{E} \succeq 0 \implies (\boldsymbol{D} \otimes \boldsymbol{E}) \succeq 0; \quad \boldsymbol{D} \succ 0, \boldsymbol{E} \succ 0 \implies (\boldsymbol{D} \otimes \boldsymbol{E}) \succ 0. \tag{37d}$$

Proofs of (36) and (37) can be found in (Matsaglia & PH Styan (1974)) and (Magnus & Neudecker (1986)), respectively.

To derive the results presented in corollary (4.1.1.1), we use the following definitions of the Moore-Penrose inverse of a matrix (denoted by $(\cdot)^+$). For any matrix $\boldsymbol{D}$ and its (full) SVD, i.e., $\boldsymbol{D} = \boldsymbol{U_D} \boldsymbol{\Sigma_D} \boldsymbol{V_D}^\top$,

$$\boldsymbol{D}^+ = (\boldsymbol{D}^\top \boldsymbol{D})^{-1} \boldsymbol{D}^\top, \quad \text{when } (\boldsymbol{D}^\top \boldsymbol{D})^{-1} \text{ exists}, \tag{38a}$$

$$\boldsymbol{D}^+ = \boldsymbol{D}^\top (\boldsymbol{D}\boldsymbol{D}^\top)^{-1}, \quad \text{when } (\boldsymbol{D}\boldsymbol{D}^\top)^{-1} \text{ exists}, \tag{38b}$$

$$\boldsymbol{D}^+ = \boldsymbol{V_D} \boldsymbol{\Sigma_D}^+ \boldsymbol{U_D}^\top, \tag{38c}$$

$$\boldsymbol{D}^+ = \lim_{\varepsilon \to 0} (\boldsymbol{D}^\top \boldsymbol{D} + \varepsilon^2 \boldsymbol{I})^{-1} \boldsymbol{D}^\top = \lim_{\varepsilon \to 0} \boldsymbol{D}^\top (\boldsymbol{D}\boldsymbol{D}^\top + \varepsilon^2 \boldsymbol{I})^{-1}, \tag{38d}$$

where $\boldsymbol{I}$ is the identity matrix of compatible dimension. The proof of (38d) can be found in (Albert (1972)).

To prove Theorem 4.1.1, we use some well-known results, summarized as the following lemma in (Baldi & Hornik (1989)), for linear least-squares optimization.

**Lemma A.0.1.** *The quadratic function $L(\boldsymbol{z}) = \|\boldsymbol{y} - \boldsymbol{M}\boldsymbol{z}\|^2 = \boldsymbol{y}^\top \boldsymbol{y} - 2\boldsymbol{y}^\top \boldsymbol{M}\boldsymbol{z} + \boldsymbol{z}^\top \boldsymbol{M}^\top \boldsymbol{M}\boldsymbol{z}$ is convex, and a point $\boldsymbol{z}$ globally minimizes $L$ if and only if $\nabla L(\boldsymbol{z}) = 0$, or equivalently, $\boldsymbol{M}^\top \boldsymbol{M}\boldsymbol{z} = \boldsymbol{M}^\top \boldsymbol{y}$. Furthermore, if $\boldsymbol{M}^\top \boldsymbol{M} \succ 0$, i.e., positive definite, then $L$ is strictly convex and reaches its unique minimum for $\boldsymbol{z} = (\boldsymbol{M}^\top \boldsymbol{M})^{-1} \boldsymbol{M}^\top \boldsymbol{y}$.*

### A.1   Proof of theorem 4.1.1

**Theorem 4.1.1.** *Consider the following objective function*

$$L_{\text{pred}}(\boldsymbol{E_x}, \boldsymbol{G}) = \frac{1}{n} \sum_{i=0}^{n-1} \left\| \boldsymbol{E_x}\boldsymbol{x}(t_{i+1}) - \boldsymbol{G}\boldsymbol{E_{xu}}\boldsymbol{\omega}(t_i) \right\|^2, \tag{12}$$

where $G = [A_R \quad B_R] \in \mathbb{R}^{r_x \times (r_x + d_u)}$, $E_{xu} = \begin{bmatrix} E_x & 0 \\ 0 & I_{d_u} \end{bmatrix} \in \mathbb{R}^{(r_x + d_u) \times (d_x + d_u)}$, $I_{d_u}$ being the identity matrix of order $d_u$. For any fixed matrix $E_x$, the objective function $L_{\text{pred}}$ is convex in the coefficients of $G$ and attains its minimum for any $G$ satisfying

$$GE_{xu}\Omega\Omega^\top E_{xu}^\top = E_x Y \Omega^\top E_{xu}^\top, \tag{13}$$

where $Y$ and $\Omega$ are the data matrices as defined in section (3.3). If $E_x$ has full rank $r_x$, and $\Omega\Omega^\top$ is non-singular, then $L_{\text{pred}}$ is strictly convex and has a unique minimum for

$$G = [A_R \quad B_R] = E_x Y \Omega^\top E_{xu}^\top (E_{xu}\Omega\Omega^\top E_{xu}^\top)^{-1}. \tag{14}$$

*Proof.* We can write $L_{\text{pred}}(E_x, G)$ as follows,

$$\begin{aligned} L_{\text{pred}}(E_x, G) &= \frac{1}{n} \sum_{i=0}^{n-1} \left\| E_x x(t_{i+1}) - GE_{xu}\omega(t_i) \right\|^2 \\ &= \left\| \text{vec}(E_x Y) - \text{vec}(GE_{xu}\Omega) \right\|^2 \\ &= \left\| \text{vec}(E_x Y) - (\Omega^\top E_{xu}^\top \otimes I_{r_x})\text{vec}(G) \right\|^2. \end{aligned} \tag{39}$$

The third equality is obtained using (37a). For fixed $E_x$, we can apply Lemma A.0.1 to (39): (39) is convex in coefficient of $G$, and $G$ corresponds to a global minimum of $L_{\text{pred}}$ if and only if

$$(\Omega^\top E_{xu}^\top \otimes I_{r_x})^\top (\Omega^\top E_{xu}^\top \otimes I_{r_x})\text{vec}(G) = (\Omega^\top E_{xu}^\top \otimes I_{r_x})^\top \text{vec}(E_x Y). \tag{40}$$

Using (37b) and (37c), we can write (40) as

$$(E_{xu}\Omega\Omega^\top E_{xu}^\top \otimes I_{r_x})\text{vec}(G) = (E_{xu}\Omega \otimes I_{r_x})\text{vec}(E_x Y). \tag{41}$$

Applying (37a) on (41), we get $GE_{xu}\Omega\Omega^\top E_{xu}^\top = E_x Y \Omega^\top E_{xu}^\top$, i.e., (13).

If $E_x$ has full rank $r_x$, then $E_{xu} = \begin{bmatrix} E_x & 0 \\ 0 & I_{d_u} \end{bmatrix} \in \mathbb{R}^{(r_x + d_u) \times (d_x + d_u)}$ has full rank $(r_x + d_u)$. If $\Omega\Omega^\top \in \mathbb{R}^{(d_x + d_u) \times (d_x + d_u)}$ is non-singular, then $\Omega$ has full row-rank $(d_x + d_u)$. Consequently, using (36a) and (36b), we have

$$\text{rank}(E_{xu}\Omega\Omega^\top E_{xu}^\top) = \text{rank}(E_{xu}\Omega) = \text{rank}(E_{xu}) = r_x + d_u. \tag{42}$$

Hence the symmetric positive semidefinite matrix $E_{xu}\Omega\Omega^\top E_{xu}^\top$ has full rank and therefore positive definite. Using (37b), (37c), and (37d), we can see that $(\Omega^\top E_{xu}^\top \otimes I_{r_x})^\top (\Omega^\top E_{xu}^\top \otimes I_{r_x}) = (E_{xu}\Omega\Omega^\top E_{xu}^\top \otimes I_{r_x})$ is positive definite as well. Therefore, by Lemma A.0.1, (39) is strictly convex in the coefficients of $G$ and has a unique minimum. Since $E_{xu}\Omega\Omega^\top E_{xu}^\top \succ 0$, it is invertible. Hence, from (13), we can say that the unique minimum of (39) is reached at $G = E_x Y \Omega^\top E_{xu}^\top (E_{xu}\Omega\Omega^\top E_{xu}^\top)^{-1}$, i.e., (14). ∎

## A.2 An alternative representation of (14)

Here we provide a possible alternative representation of (14) required to prove corollary 4.1.1.1.

**Lemma A.2.1.** *Consider the (full) SVD of the data matrix $\Omega$ given by $\Omega = U_\Omega \Sigma_\Omega V_\Omega^\top$, where $U_\Omega \in \mathbb{R}^{(d_x + d_u) \times (d_x + d_u)}$, $\Sigma_\Omega \in \mathbb{R}^{(d_x + d_u) \times n}$, and $V_\Omega \in \mathbb{R}^{n \times n}$. (14) can be expressed as*

$$G = \lim_{\varepsilon \to 0} E_x Y V_\Omega (\Sigma_\Omega^\top U_\Omega^\top E_{xu}^\top E_{xu} U_\Omega \Sigma_\Omega + \varepsilon^2 I_n)^{-1} \Sigma_\Omega^\top U_\Omega^\top E_{xu}^\top. \tag{43}$$

*Proof.* Replacing $\Omega$ with its SVD in (14) we get,

$$\begin{aligned} G &= E_x Y V_\Omega \Sigma_\Omega^\top U_\Omega^\top E_{xu}^\top (E_{xu} U_\Omega \Sigma_\Omega V_\Omega^\top V_\Omega \Sigma_\Omega^\top U_\Omega^\top E_{xu}^\top)^{-1} \\ &= E_x Y V_\Omega \Sigma_\Omega^\top U_\Omega^\top E_{xu}^\top (E_{xu} U_\Omega \Sigma_\Omega \Sigma_\Omega^\top U_\Omega^\top E_{xu}^\top)^{-1} \\ &= E_x Y V_\Omega (E_{xu} U_\Omega \Sigma_\Omega)^+ \end{aligned} \tag{44}$$

The second equality is due to the orthogonality of $\boldsymbol{V_\Omega}$. The third equality is obtained using (38b). Substituting $(\boldsymbol{E_{xu}U_\Omega\Sigma_\Omega})^+$ with the *limit definition* (38d) of the Moore-Penrose inverse, we get

$$\boldsymbol{G} = \lim_{\varepsilon \to 0} \boldsymbol{E_x Y V_\Omega}(\boldsymbol{\Sigma_\Omega^\top U_\Omega^\top E_{xu}^\top E_{xu} U_\Omega \Sigma_\Omega} + \varepsilon^2 \boldsymbol{I}_n)^{-1}\boldsymbol{\Sigma_\Omega^\top U_\Omega^\top E_{xu}^\top}. \tag{45}$$

∎

### A.3  Proof of Corollary 4.1.1.1

**Corollary 4.1.1.1.** *Consider the (full) SVD of the data matrix $\boldsymbol{\Omega}$ given by $\boldsymbol{\Omega} = \boldsymbol{U_\Omega \Sigma_\Omega V_\Omega^\top}$, where $\boldsymbol{U_\Omega} \in \mathbb{R}^{(d_x+d_u)\times(d_x+d_u)}$, $\boldsymbol{\Sigma_\Omega} \in \mathbb{R}^{(d_x+d_u)\times n}$, and $\boldsymbol{V_\Omega} \in \mathbb{R}^{n\times n}$. If $\boldsymbol{E_x} = \widehat{\boldsymbol{U}}_Y^\top$ and $\boldsymbol{\Omega\Omega}^\top$ is non-singular, then the solution for $\boldsymbol{G} = [\boldsymbol{A}_R \quad \boldsymbol{B}_R]$ corresponding to the unique minimum of $L_{pred}$ can be expressed as*

$$\boldsymbol{A}_R = \widehat{\boldsymbol{U}}_Y^\top \boldsymbol{Y V_\Omega} \Sigma^* \boldsymbol{U}_{\Omega,1}^\top \widehat{\boldsymbol{U}}_Y, \quad and \quad \boldsymbol{B}_R = \widehat{\boldsymbol{U}}_Y^\top \boldsymbol{Y V_\Omega} \Sigma^* \boldsymbol{U}_{\Omega,2}^\top, \tag{15}$$

*where $[\boldsymbol{U}_{\Omega,1}^\top \quad \boldsymbol{U}_{\Omega,2}^\top] = \boldsymbol{U}_\Omega^\top$ with $\boldsymbol{U}_{\Omega,1} \in \mathbb{R}^{d_x\times(d_x+d_u)}, \boldsymbol{U}_{\Omega,2} \in \mathbb{R}^{d_u\times(d_x+d_u)}$, and*
$\Sigma^* = \lim_{\varepsilon\to 0}(\boldsymbol{\Sigma_\Omega^\top U}_{\Omega,1}^\top \widehat{\boldsymbol{U}}_Y \widehat{\boldsymbol{U}}_Y^\top \boldsymbol{U}_{\Omega,1} \boldsymbol{\Sigma_\Omega} + \boldsymbol{\Sigma_\Omega^\top U}_{\Omega,2}^\top \boldsymbol{U}_{\Omega,2} \boldsymbol{\Sigma_\Omega} + \varepsilon^2 \boldsymbol{I}_n)^{-1}\boldsymbol{\Sigma_\Omega^\top}.$

*Proof.* By the definition of truncated SVD, the columns of $\widehat{\boldsymbol{U}}_Y$ are orthonormal. Therefore, $\widehat{\boldsymbol{U}}_Y^\top$ has full row-rank $r_x$. Hence, by theorem 4.1.1 and lemma A.2.1, if $\boldsymbol{E_x} = \widehat{\boldsymbol{U}}_Y^\top$, and $\boldsymbol{\Omega\Omega}^\top$ is non-singular, then the unique minimum of $L_{\text{pred}}$, is reached when

$$\boldsymbol{G} = \widehat{\boldsymbol{U}}_Y^\top \boldsymbol{Y V_\Omega}(\boldsymbol{E_{xu}U_\Omega\Sigma_\Omega})^+ = \lim_{\varepsilon\to 0}\widehat{\boldsymbol{U}}_Y^\top \boldsymbol{Y V_\Omega}(\boldsymbol{\Sigma_\Omega^\top U_\Omega^\top E_{xu}^\top E_{xu}U_\Omega\Sigma_\Omega} + \varepsilon^2 \boldsymbol{I}_n)^{-1}\boldsymbol{\Sigma_\Omega^\top U_\Omega^\top E_{xu}^\top}. \tag{46}$$

Now, substituting $\boldsymbol{E_x} = \widehat{\boldsymbol{U}}_Y^\top$ in $\boldsymbol{E_{xu}}$, and using the partition $\boldsymbol{U}_\Omega^\top = [\boldsymbol{U}_{\Omega,1}^\top \quad \boldsymbol{U}_{\Omega,2}^\top]$, where $\boldsymbol{U}_{\Omega,1} \in \mathbb{R}^{d_x\times(d_x+d_u)}, \boldsymbol{U}_{\Omega,2} \in \mathbb{R}^{d_u\times(d_x+d_u)}$, we get

$$\boldsymbol{E_{xu}U_\Omega} = \begin{bmatrix} \widehat{\boldsymbol{U}}_Y^\top & \boldsymbol{0} \\ \boldsymbol{0} & \boldsymbol{I}_{d_u} \end{bmatrix} \begin{bmatrix} \boldsymbol{U}_{\Omega,1} \\ \boldsymbol{U}_{\Omega,2} \end{bmatrix} = \begin{bmatrix} \widehat{\boldsymbol{U}}_Y^\top \boldsymbol{U}_{\Omega,1} \\ \boldsymbol{U}_{\Omega,2} \end{bmatrix}, \tag{47}$$

and

$$\boldsymbol{U}_\Omega^\top \boldsymbol{E_{xu}^\top E_{xu} U_\Omega} = \begin{bmatrix} \boldsymbol{U}_{\Omega,1}^\top \widehat{\boldsymbol{U}}_Y & \boldsymbol{U}_{\Omega,2}^\top \end{bmatrix} \begin{bmatrix} \widehat{\boldsymbol{U}}_Y^\top \boldsymbol{U}_{\Omega,1} \\ \boldsymbol{U}_{\Omega,2} \end{bmatrix} = \boldsymbol{U}_{\Omega,1}^\top \widehat{\boldsymbol{U}}_Y \widehat{\boldsymbol{U}}_Y^\top \boldsymbol{U}_{\Omega,1} + \boldsymbol{U}_{\Omega,2}^\top \boldsymbol{U}_{\Omega,2}. \tag{48}$$

Plugging (47) and (48) into (46) leads to

$$\boldsymbol{G} = \lim_{\varepsilon\to 0}\widehat{\boldsymbol{U}}_Y^\top \boldsymbol{Y V_\Omega}(\boldsymbol{\Sigma_\Omega^\top U}_{\Omega,1}^\top \widehat{\boldsymbol{U}}_Y \widehat{\boldsymbol{U}}_Y^\top \boldsymbol{U}_{\Omega,1} \boldsymbol{\Sigma_\Omega} + \boldsymbol{\Sigma_\Omega^\top U}_{\Omega,2}^\top \boldsymbol{U}_{\Omega,2} \boldsymbol{\Sigma_\Omega} + \varepsilon^2 \boldsymbol{I}_n)^{-1}\boldsymbol{\Sigma_\Omega^\top} \begin{bmatrix} \boldsymbol{U}_{\Omega,1}^\top \widehat{\boldsymbol{U}}_Y & \boldsymbol{U}_{\Omega,2}^\top \end{bmatrix}. \tag{49}$$

Defining $\Sigma^* \triangleq \lim_{\varepsilon\to 0}(\boldsymbol{\Sigma_\Omega^\top U}_{\Omega,1}^\top \widehat{\boldsymbol{U}}_Y \widehat{\boldsymbol{U}}_Y^\top \boldsymbol{U}_{\Omega,1} \boldsymbol{\Sigma_\Omega} + \boldsymbol{\Sigma_\Omega^\top U}_{\Omega,2}^\top \boldsymbol{U}_{\Omega,2} \boldsymbol{\Sigma_\Omega} + \varepsilon^2 \boldsymbol{I}_n)^{-1}\boldsymbol{\Sigma_\Omega^\top}$, we can split (49) into

$$\boldsymbol{A}_R = \widehat{\boldsymbol{U}}_Y^\top \boldsymbol{Y V_\Omega} \Sigma^* \boldsymbol{U}_{\Omega,1}^\top \widehat{\boldsymbol{U}}_Y, \quad and \quad \boldsymbol{B}_R = \widehat{\boldsymbol{U}}_Y^\top \boldsymbol{Y V_\Omega} \Sigma^* \boldsymbol{U}_{\Omega,2}^\top,$$

which is (15). ∎

### A.4  The case when $\boldsymbol{\Omega\Omega}^\top$ not invertible

When the covariance matrix $\boldsymbol{\Omega\Omega}^\top$ is not invertible, which is always true if $n < d_x + d_u$, the matrix $\boldsymbol{E_{xu}\Omega\Omega}^\top \boldsymbol{E_{xu}^\top}$ is not guaranteed to be invertible. In that case, the minimum of $L_{\text{pred}}$ corresponds to infinitely many solutions for $\boldsymbol{G}$. However, minimizing $L_{\text{pred}}$ with added $\ell_2$ regularization, i.e., $L_{\text{pred,reg}}(\boldsymbol{E_x}, \boldsymbol{G}) = L_{\text{pred}}(\boldsymbol{E_x}, \boldsymbol{G}) + \beta\|\text{vec}(\boldsymbol{G})\|^2$ provides a unique solution for $\boldsymbol{G}$, for a fixed $\boldsymbol{E_x}$. We have the following result.

**Theorem A.4.1.** *For any fixed matrix $\boldsymbol{E_x}$ and $\beta > 0$, the objective function $L_{\text{pred,reg}}(\boldsymbol{E_x}, \boldsymbol{G}) = L_{\text{pred}}(\boldsymbol{E_x}, \boldsymbol{G}) + \beta\|vec(\boldsymbol{G})\|^2$ is strictly convex in the coefficients of $\boldsymbol{G}$, and the global minimum of $L_{\text{pred,reg}}$ corresponds to the unique solution for $\boldsymbol{G}$, given by*

$$\boldsymbol{G} = \boldsymbol{E_x Y \Omega}^\top \boldsymbol{E_{xu}^\top}(\boldsymbol{E_{xu}\Omega\Omega}^\top \boldsymbol{E_{xu}^\top} + \beta \boldsymbol{I}_{r_x+d_u})^{-1}. \tag{50}$$

*Proof.* $L_{\text{pred,reg}}(\boldsymbol{E_x}, \boldsymbol{G})$ can be written as, using (37a-c),

$$
\begin{aligned}
L_{\text{pred,reg}}(\boldsymbol{E_x}, \boldsymbol{G}) &= \left\| \text{vec}(\boldsymbol{E_x Y}) - (\boldsymbol{\Omega}^\top \boldsymbol{E_{xu}}^\top \otimes \boldsymbol{I}_{r_x}) \text{vec}(\boldsymbol{G}) \right\|^2 + \beta \|\text{vec}(\boldsymbol{G})\|^2 \\
&= \text{vec}(\boldsymbol{E_x Y})^\top \text{vec}(\boldsymbol{E_x Y}) - 2\text{vec}(\boldsymbol{E_x Y})^\top (\boldsymbol{\Omega}^\top \boldsymbol{E_{xu}}^\top \otimes \boldsymbol{I}_{r_x}) \text{vec}(\boldsymbol{G}) \\
&\quad + \text{vec}(\boldsymbol{G})^\top (\boldsymbol{E_{xu} \Omega \Omega}^\top \boldsymbol{E_{xu}}^\top \otimes \boldsymbol{I}_{r_x} + \beta \boldsymbol{I}_{r_x(r_x+d_u)}) \text{vec}(\boldsymbol{G})
\end{aligned}
$$

$\boldsymbol{E_{xu} \Omega \Omega}^\top \boldsymbol{E_{xu}}^\top$ is a symmetric positive semidefinite matrix, irrespective of whether it has full rank or not. Hence, by (37d), $\boldsymbol{E_{xu} \Omega \Omega}^\top \boldsymbol{E_{xu}}^\top \otimes \boldsymbol{I}_{r_x}$ is symmetric positive semidefinite. Consequently, for any $\beta > 0$, $\boldsymbol{E_{xu} \Omega \Omega}^\top \boldsymbol{E_{xu}}^\top \otimes \boldsymbol{I}_{r_x} + \beta \boldsymbol{I}_{r_x(r_x+d_u)}$ is positive definite. According to lemma A.0.1, $L_{\text{pred,reg}}$ is therefore strictly convex in the coefficients of $\boldsymbol{G}$ and globally minimized when $\nabla L_{\text{pred,reg}} = 0$. The unique solution of (50) can be derived in the same manner as theorem 4.1.1. ∎

**Remark.** Replacing $\boldsymbol{\Omega}$ with its SVD in (50) we get,

$$
\boldsymbol{G} = \boldsymbol{E_x Y V_\Omega \Sigma_\Omega}^\top \boldsymbol{U_\Omega}^\top \boldsymbol{E_{xu}}^\top (\boldsymbol{E_{xu} U_\Omega \Sigma_\Omega \Sigma_\Omega}^\top \boldsymbol{U_\Omega}^\top \boldsymbol{E_{xu}}^\top + \beta \boldsymbol{I}_{r_x+d_u})^{-1}. \tag{51}
$$

In the limit $\beta \to 0^+$, (51) converges to (44).

## A.5 DMDc through a linear autoencoding structure

Here we present a linear autoencoding structure that leads to a linear ROM exactly resembling the DMDc solution when $\boldsymbol{E_x} = \widehat{\boldsymbol{U}}_{\boldsymbol{Y}}^\top$. However, its DNN-based nonlinear counterpart does not actually offer dimensionality reduction.

**Theorem A.5.1.** *Consider the following objective function*

$$
L_{\text{pred,alt}}(\boldsymbol{E_x}, \widetilde{\boldsymbol{G}}) = \frac{1}{n} \sum_{i=0}^{n-1} \left\| \boldsymbol{E_x x}(t_{i+1}) - \widetilde{\boldsymbol{G}} \boldsymbol{\omega}(t_i) \right\|^2, \tag{52}
$$

*where $\widetilde{\boldsymbol{G}} \in \mathbb{R}^{r_x \times (d_x+d_u)}$. For any fixed matrix $\boldsymbol{E_x}$, the objective function $L_{\text{pred,alt}}$ is convex in the coefficients of $\widetilde{\boldsymbol{G}}$ and attains its minimum for any $\widetilde{\boldsymbol{G}}$ satisfying*

$$
\widetilde{\boldsymbol{G}} \boldsymbol{\Omega \Omega}^\top = \boldsymbol{E_x Y \Omega}^\top, \tag{53}
$$

*where $\boldsymbol{Y}$ and $\boldsymbol{\Omega}$ are the data matrices as defined in section (3.3). If $\boldsymbol{\Omega \Omega}^\top$ is non-singular, then $L_{\text{pred,alt}}$ is strictly convex and has a unique minimum for*

$$
\widetilde{\boldsymbol{G}} = \boldsymbol{E_x Y \Omega}^\top (\boldsymbol{\Omega \Omega}^\top)^{-1}. \tag{54}
$$

*Proof.* The proof is very similar to the proof of theorem 4.1.1. Using (37a), we can write $L_{\text{pred,alt}}(\boldsymbol{E_x}, \widetilde{\boldsymbol{G}})$ as follows,

$$
\begin{aligned}
L_{\text{pred,alt}}(\boldsymbol{E_x}, \widetilde{\boldsymbol{G}}) &= \frac{1}{n} \sum_{i=0}^{n-1} \left\| \boldsymbol{E_x x}(t_{i+1}) - \widetilde{\boldsymbol{G}} \boldsymbol{\omega}(t_i) \right\|^2 \\
&= \left\| \text{vec}(\boldsymbol{E_x Y}) - \text{vec}(\widetilde{\boldsymbol{G}} \boldsymbol{\Omega}) \right\|^2 \\
&= \left\| \text{vec}(\boldsymbol{E_x Y}) - (\boldsymbol{\Omega}^\top \otimes \boldsymbol{I}_{r_x}) \text{vec}(\widetilde{\boldsymbol{G}}) \right\|^2. \tag{55}
\end{aligned}
$$

For fixed $\boldsymbol{E_x}$, applying Lemma A.0.1 to (55), we can say $L_{\text{pred,alt}}$ is convex in the coefficients of $\widetilde{\boldsymbol{G}}$, and $\widetilde{\boldsymbol{G}}$ corresponds to a global minimum of $L_{\text{pred,alt}}$ if and only if

$$
(\boldsymbol{\Omega}^\top \otimes \boldsymbol{I}_{r_x})^\top (\boldsymbol{\Omega}^\top \otimes \boldsymbol{I}_{r_x}) \text{vec}(\widetilde{\boldsymbol{G}}) = (\boldsymbol{\Omega}^\top \otimes \boldsymbol{I}_{r_x})^\top \text{vec}(\boldsymbol{E_x Y}). \tag{56}
$$

Using (37a-c), we can write (56) as $\widetilde{\boldsymbol{G}} \boldsymbol{\Omega \Omega}^\top = \boldsymbol{E_x Y \Omega}^\top$, which is (53).

If $\boldsymbol{\Omega\Omega}^\top$ is non-singular, then it is symmetric positive definite. Using (37b-d), we can see that $(\boldsymbol{\Omega}^\top \otimes \boldsymbol{I}_{r_{\boldsymbol{x}}})^\top (\boldsymbol{\Omega}^\top \otimes \boldsymbol{I}_{r_{\boldsymbol{x}}}) = (\boldsymbol{\Omega\Omega}^\top \otimes \boldsymbol{I}_{r_{\boldsymbol{x}}})$ is positive definite as well. Therefore, by Lemma A.0.1, (55) is strictly convex in coefficient in $\widetilde{\boldsymbol{G}}$ and has a unique minimum. In that case, from (53), we can say that the unique minimum of (55) is reached at $\widetilde{\boldsymbol{G}} = \boldsymbol{E}_{\boldsymbol{x}} \boldsymbol{Y} \boldsymbol{\Omega}^\top (\boldsymbol{\Omega\Omega}^\top)^{-1}$, i.e., (54). ∎

**Corollary A.5.1.1.** *Consider the (full) SVD of the data matrix $\boldsymbol{\Omega}$ given by $\boldsymbol{\Omega} = \boldsymbol{U_\Omega \Sigma_\Omega V_\Omega}^\top$, where $\boldsymbol{U_\Omega} \in \mathbb{R}^{(d_{\boldsymbol{x}}+d_{\boldsymbol{u}})\times(d_{\boldsymbol{x}}+d_{\boldsymbol{u}})}$, $\boldsymbol{\Sigma_\Omega} \in \mathbb{R}^{(d_{\boldsymbol{x}}+d_{\boldsymbol{u}})\times n}$, and $\boldsymbol{V_\Omega} \in \mathbb{R}^{n\times n}$. If $\boldsymbol{E}_{\boldsymbol{x}} = \widehat{\boldsymbol{U}}_{\boldsymbol{Y}}^\top$ and $\boldsymbol{\Omega\Omega}^\top$ is non-singular, then the solution for $\widetilde{\boldsymbol{G}}$ corresponding to the unique minimum of $L_{\mathrm{pred,alt}}$ can be expressed as*

$$\widetilde{\boldsymbol{G}} = \widehat{\boldsymbol{U}}_{\boldsymbol{Y}}^\top \boldsymbol{Y} \boldsymbol{V_\Omega} \Sigma_{\boldsymbol{\Omega}}^+ \boldsymbol{U_\Omega}^\top. \tag{57}$$

*Proof.* By theorem A.5.1, if $\boldsymbol{E}_{\boldsymbol{x}} = \widehat{\boldsymbol{U}}_{\boldsymbol{Y}}^\top$, and $\boldsymbol{\Omega\Omega}^\top$ is non-singular, then the unique minimum of $L_{\mathrm{pred,alt}}$ is reached when

$$\widetilde{\boldsymbol{G}} = \widehat{\boldsymbol{U}}_{\boldsymbol{Y}}^\top \boldsymbol{Y} \boldsymbol{\Omega}^\top (\boldsymbol{\Omega\Omega}^\top)^{-1} = \widehat{\boldsymbol{U}}_{\boldsymbol{Y}}^\top \boldsymbol{Y} \boldsymbol{\Omega}^+ \tag{58}$$

The second equality is due to (38b). Substituting $\boldsymbol{\Omega}^+$ with its *SVD definition* (38c) into (58), we get $\widehat{\boldsymbol{U}}_{\boldsymbol{Y}}^\top \boldsymbol{Y} \boldsymbol{V_\Omega} \Sigma_{\boldsymbol{\Omega}}^+ \boldsymbol{U_\Omega}^\top$, which is (57). ∎

**Remark.** From (52), it can be seen that $\widetilde{\boldsymbol{G}}$ maps the concatenated vector, $\boldsymbol{\omega}(t_i)$, of full state and actuation to the next reduce state $\boldsymbol{x}_\mathrm{R}(t_{i+1})$. We can partition (57) as $\widetilde{\boldsymbol{G}} = \widehat{\boldsymbol{U}}_{\boldsymbol{Y}}^\top \boldsymbol{Y} \boldsymbol{V_\Omega} \Sigma_{\boldsymbol{\Omega}}^+ [\boldsymbol{U}_{\boldsymbol{\Omega},1}^\top \quad \boldsymbol{U}_{\boldsymbol{\Omega},2}^\top] = [\widetilde{\boldsymbol{A}} \quad \widetilde{\boldsymbol{B}}]$ to separate out the blocks corresponding to state and actuation. Here, $\boldsymbol{U}_{\boldsymbol{\Omega},1}, \boldsymbol{U}_{\boldsymbol{\Omega},2}$ are the same as defined in corollary 4.1.1.1, and $\widetilde{\boldsymbol{A}} \in \mathbb{R}^{r_{\boldsymbol{x}}\times d_{\boldsymbol{x}}}, \widetilde{\boldsymbol{B}} \in \mathbb{R}^{r_{\boldsymbol{x}}\times d_{\boldsymbol{u}}}$. Now, if we post-multiply $\widetilde{\boldsymbol{A}}$ with $\boldsymbol{E}_{\boldsymbol{x}}^\top = \widehat{\boldsymbol{U}}_{\boldsymbol{Y}} \in \mathbb{R}^{d_{\boldsymbol{x}}\times r_{\boldsymbol{x}}}$, we get a ROM

$$\widetilde{\boldsymbol{A}}_\mathrm{R} = \widetilde{\boldsymbol{A}}\widehat{\boldsymbol{U}}_{\boldsymbol{Y}} = \widehat{\boldsymbol{U}}_{\boldsymbol{Y}}^\top \boldsymbol{Y} \boldsymbol{V_\Omega} \Sigma_{\boldsymbol{\Omega}}^+ \boldsymbol{U}_{\boldsymbol{\Omega},1}^\top \widehat{\boldsymbol{U}}_{\boldsymbol{Y}}, \qquad \widetilde{\boldsymbol{B}}_\mathrm{R} = \widetilde{\boldsymbol{B}} = \widehat{\boldsymbol{U}}_{\boldsymbol{Y}}^\top \boldsymbol{Y} \boldsymbol{V_\Omega} \Sigma_{\boldsymbol{\Omega}}^+ \boldsymbol{U}_{\boldsymbol{\Omega},2}^\top, \tag{59}$$

which maps the current reduced state $\boldsymbol{x}_\mathrm{R}(t_i)$ and actuation $\boldsymbol{u}(t_i)$ to the next reduced state $\boldsymbol{x}_\mathrm{R}(t_{i+1})$. It can be verified easily that if we use the truncated SVD (as defined by 9), instead of the full SVD, for $\boldsymbol{\Omega}$ in (58) and follow the similar steps afterward, we get an approximation of (59):

$$\widehat{\boldsymbol{A}}_\mathrm{R} = \widehat{\boldsymbol{U}}_{\boldsymbol{Y}}^\top \boldsymbol{Y} \widehat{\boldsymbol{V}}_{\boldsymbol{\Omega}} \widehat{\Sigma}_{\boldsymbol{\Omega}}^{-1} \widehat{\boldsymbol{U}}_{\boldsymbol{\Omega},1}^\top \widehat{\boldsymbol{U}}_{\boldsymbol{Y}} = \boldsymbol{A}_\mathrm{R,DMDc}; \qquad \widehat{\boldsymbol{B}}_\mathrm{R} = \widehat{\boldsymbol{U}}_{\boldsymbol{Y}}^\top \boldsymbol{Y} \widehat{\boldsymbol{V}}_{\boldsymbol{\Omega}} \widehat{\Sigma}_{\boldsymbol{\Omega}}^{-1} \widehat{\boldsymbol{U}}_{\boldsymbol{\Omega},2}^\top = \boldsymbol{B}_\mathrm{R,DMDc}.$$

In summary, the aforementioned method can be carried out using gradient descent-based optimization and leads to the same ROM as DMDc, when $\boldsymbol{E}_{\boldsymbol{x}} = \widehat{\boldsymbol{U}}_{\boldsymbol{Y}}^\top$. However, in this method, the benefit of dimensionality reduction is realized only when linear networks are used. A nonlinear counterpart (a DNN in the context of this paper) of $\widetilde{\boldsymbol{A}}_\mathrm{R}$, i.e., a nonlinear mapping from $\mathbb{R}^{r_{\boldsymbol{x}}}$ to $\mathbb{R}^{r_{\boldsymbol{x}}}$, cannot be pre-computed from a nonlinear counterpart of $\widetilde{\boldsymbol{G}}$, unlike the linear case (59). Consequently, we lose the benefit of dimensionality reduction when nonlinear networks are used.

## A.6 Proof of theorem 4.2.1

**Theorem 4.2.1.** *Consider the target dynamics defined by (28)and the candidate Lyapunov function defined by (29). Suppose the difference between the target dynamics and the closed-loop dynamics satisfies*

$$\|\mathcal{F}(\boldsymbol{x}_\mathrm{R}, \mathcal{E}_{\boldsymbol{u}} \circ \Pi(\boldsymbol{x}_\mathrm{R})) - \mathcal{F}_s(\boldsymbol{x}_\mathrm{R})\| \le \delta < \frac{\alpha\theta\lambda_{\min}(\boldsymbol{K})}{2\lambda_{\max}(\boldsymbol{K})}\sqrt{\frac{\lambda_{\min}(\boldsymbol{K})}{\lambda_{\max}(\boldsymbol{K})}}\eta, \tag{30}$$

*for all $\boldsymbol{x}_\mathrm{R} \in \mathbb{X}_\mathrm{R} = \{\boldsymbol{x}_\mathrm{R} \in \mathbb{R}^{r_{\boldsymbol{x}}} \mid \|\boldsymbol{x}_\mathrm{R}\| < \eta\}$ and $0 < \theta < 1$. Then, for all initial points satisfying $\|\boldsymbol{x}_\mathrm{R}(t_0)\| < \sqrt{\frac{\lambda_{\min}(\boldsymbol{K})}{\lambda_{\max}(\boldsymbol{K})}}\eta$, the solution of the closed-loop ROM $\frac{d\boldsymbol{x}_\mathrm{R}}{dt} = \mathcal{F}(\boldsymbol{x}_\mathrm{R}, \mathcal{E}_{\boldsymbol{u}} \circ \Pi(\boldsymbol{x}_\mathrm{R}))$ satisfies*

$$\|\boldsymbol{x}_\mathrm{R}(t)\| \le \lambda e^{-\gamma(t-t_0)}\|\boldsymbol{x}_\mathrm{R}(t_0)\|, \quad \forall t_0 \le t < t_c + t_0 \tag{31}$$

*and*

$$\|\boldsymbol{x}_\mathrm{R}(t)\| \le \frac{2\delta}{\alpha\theta}\lambda^3, \quad \forall t \ge t_c + t_0 \tag{32}$$

*for some finite $t_c > 0$, where*

$$\gamma = \frac{\alpha(1-\theta)\lambda_{\min}(\boldsymbol{K})}{2\lambda_{\max}(\boldsymbol{K})} \quad and \quad \lambda = \sqrt{\frac{\lambda_{\max}(\boldsymbol{K})}{\lambda_{\min}(\boldsymbol{K})}} \tag{33}$$

*Proof.* From the definition of $\mathcal{V}_{\mathrm{R}}$, we have

$$\lambda_{\min}(\boldsymbol{K})\|\boldsymbol{x}_{\mathrm{R}}\|^2 \le \mathcal{V}_{\mathrm{R}}(\boldsymbol{x}_{\mathrm{R}}) \le \lambda_{\max}(\boldsymbol{K})\|\boldsymbol{x}_{\mathrm{R}}\|^2, \quad \forall\, \boldsymbol{x}_{\mathrm{R}} \in \mathbb{R}^{r_x}, \tag{60}$$

where $\lambda_{\min}(\boldsymbol{K})$ and $\lambda_{\max}(\boldsymbol{K})$ denote the smallest and largest eigenvalues, respectively, of $\boldsymbol{K}$ and have positive values since the matrix $\boldsymbol{K}$ is positive definite. Moreover, the definition of the target dynamics (28) implies

$$
\begin{aligned}
\nabla\mathcal{V}_{\mathrm{R}}(\boldsymbol{x}_{\mathrm{R}})^\top \mathcal{F}_s(\boldsymbol{x}_{\mathrm{R}}) &= \begin{cases} \nabla\mathcal{V}_{\mathrm{R}}(\boldsymbol{x}_{\mathrm{R}})^\top \mathcal{P}(\boldsymbol{x}_{\mathrm{R}}), & \text{if } \nabla\mathcal{V}_{\mathrm{R}}(\boldsymbol{x}_{\mathrm{R}})^\top \mathcal{P}(\boldsymbol{x}_{\mathrm{R}}) \le -\alpha\mathcal{V}_{\mathrm{R}}(\boldsymbol{x}_{\mathrm{R}}) \\ \nabla\mathcal{V}_{\mathrm{R}}(\boldsymbol{x}_{\mathrm{R}})^\top \mathcal{P}(\boldsymbol{x}_{\mathrm{R}}) - \nabla\mathcal{V}_{\mathrm{R}}(\boldsymbol{x}_{\mathrm{R}})^\top \frac{\nabla\mathcal{V}_{\mathrm{R}}(\boldsymbol{x}_{\mathrm{R}})^\top \mathcal{P}(\boldsymbol{x}_{\mathrm{R}})+\alpha\mathcal{V}_{\mathrm{R}}(\boldsymbol{x}_{\mathrm{R}})}{\|\nabla\mathcal{V}_{\mathrm{R}}(\boldsymbol{x}_{\mathrm{R}})\|^2}\nabla\mathcal{V}_{\mathrm{R}}(\boldsymbol{x}_{\mathrm{R}}), & \text{otherwise} \end{cases} \\
&= \begin{cases} \nabla\mathcal{V}_{\mathrm{R}}(\boldsymbol{x}_{\mathrm{R}})^\top \mathcal{P}(\boldsymbol{x}_{\mathrm{R}}), & \text{if } \nabla\mathcal{V}_{\mathrm{R}}(\boldsymbol{x}_{\mathrm{R}})^\top \mathcal{P}(\boldsymbol{x}_{\mathrm{R}}) \le -\alpha\mathcal{V}_{\mathrm{R}}(\boldsymbol{x}_{\mathrm{R}}) \\ -\alpha\mathcal{V}_{\mathrm{R}}(\boldsymbol{x}_{\mathrm{R}}), & \text{otherwise} \end{cases} \\
&\le -\alpha\mathcal{V}_{\mathrm{R}}(\boldsymbol{x}_{\mathrm{R}}) \\
&\le -\alpha\lambda_{\min}(\boldsymbol{K})\|\boldsymbol{x}_{\mathrm{R}}\|^2, \quad \forall\, \boldsymbol{x}_{\mathrm{R}} \in \mathbb{R}^{r_x}.
\end{aligned} \tag{61}
$$

The last inequality is due to (60).

Now, assume $\mathcal{F}(\boldsymbol{x}_{\mathrm{R}}, \mathcal{E}_{\boldsymbol{u}} \circ \Pi(\boldsymbol{x}_{\mathrm{R}})) = \mathcal{H}(\boldsymbol{x}_{\mathrm{R}}) = \mathcal{F}_s(\boldsymbol{x}_{\mathrm{R}}) + \mathcal{J}(\boldsymbol{x}_{\mathrm{R}})$ for some function $\mathcal{J} : \mathbb{R}^{r_x} \to \mathbb{R}^{r_x}$ and consider $\mathcal{V}_{\mathrm{R}}(\boldsymbol{x}_{\mathrm{R}}) = \boldsymbol{x}_{\mathrm{R}}^\top \boldsymbol{K}\boldsymbol{x}_{\mathrm{R}}$ as a candidate Lyapunov function for

$$\frac{d\boldsymbol{x}_{\mathrm{R}}}{dt} = \mathcal{H}(\boldsymbol{x}_{\mathrm{R}}) = \mathcal{F}_s(\boldsymbol{x}_{\mathrm{R}}) + \mathcal{J}(\boldsymbol{x}_{\mathrm{R}}). \tag{62}$$

We have $\|\nabla\mathcal{V}_{\mathrm{R}}(\boldsymbol{x}_{\mathrm{R}})\| = \|2\boldsymbol{K}\boldsymbol{x}_{\mathrm{R}}\| \le 2\lambda_{\max}(\boldsymbol{K})\|\boldsymbol{x}_{\mathrm{R}}\|$. The time-derivative of $\mathcal{V}_{\mathrm{R}}$ along the trajectories of (62) satisfies

$$
\begin{aligned}
\frac{d\mathcal{V}_{\mathrm{R}}}{dt} &= \nabla\mathcal{V}_{\mathrm{R}}(\boldsymbol{x}_{\mathrm{R}})^\top \mathcal{F}_s(\boldsymbol{x}_{\mathrm{R}}) + \nabla\mathcal{V}_{\mathrm{R}}(\boldsymbol{x}_{\mathrm{R}})^\top \mathcal{J}(\boldsymbol{x}_{\mathrm{R}}) \\
&\le -\alpha\lambda_{\min}(\boldsymbol{K})\|\boldsymbol{x}_{\mathrm{R}}\|^2 + \|\nabla\mathcal{V}_{\mathrm{R}}(\boldsymbol{x}_{\mathrm{R}})\|\|\mathcal{J}(\boldsymbol{x}_{\mathrm{R}})\| \\
&\le -\alpha\lambda_{\min}(\boldsymbol{K})\|\boldsymbol{x}_{\mathrm{R}}\|^2 + 2\lambda_{\max}(\boldsymbol{K})\|\boldsymbol{x}_{\mathrm{R}}\|\delta, \quad \forall\, \|\boldsymbol{x}_{\mathrm{R}}\| < \eta \\
&= -\alpha(1-\theta)\lambda_{\min}(\boldsymbol{K})\|\boldsymbol{x}_{\mathrm{R}}\|^2 - \alpha\theta\lambda_{\min}(\boldsymbol{K})\|\boldsymbol{x}_{\mathrm{R}}\|^2 + 2\lambda_{\max}(\boldsymbol{K})\|\boldsymbol{x}_{\mathrm{R}}\|\delta, \quad 0 < \theta < 1, \; \forall\, \|\boldsymbol{x}_{\mathrm{R}}\| < \eta \\
&\le -\alpha(1-\theta)\lambda_{\min}(\boldsymbol{K})\|\boldsymbol{x}_{\mathrm{R}}\|^2 < 0, \quad \text{when } \eta > \|\boldsymbol{x}_{\mathrm{R}}\| \ge \frac{2\delta\lambda_{\max}(\boldsymbol{K})}{\alpha\theta\lambda_{\min}(\boldsymbol{K})} \triangleq \mu.
\end{aligned} \tag{63}
$$

The second inequality is obtained using (61) and the third inequality is obtained using (30). Clearly, we have a non-empty region where $\frac{d\mathcal{V}_{\mathrm{R}}}{dt} < 0$ only when

$$\delta < \frac{\alpha\theta\lambda_{\min}(\boldsymbol{K})}{2\lambda_{\max}(\boldsymbol{K})}\eta. \tag{64}$$

Let $b = \lambda_{\min}(\boldsymbol{K})\eta^2$ and $c = \lambda_{\max}(\boldsymbol{K})\mu^2$. Consider the sublevel sets $\chi_b = \{\boldsymbol{x}_{\mathrm{R}} \in \mathbb{R}^{r_x} \mid \mathcal{V}_{\mathrm{R}}(\boldsymbol{x}_{\mathrm{R}}) < b\}$ and $\chi_c = \{\boldsymbol{x}_{\mathrm{R}} \in \mathbb{R}^{r_x} \mid \mathcal{V}_{\mathrm{R}}(\boldsymbol{x}_{\mathrm{R}}) \le c\}$. It can be easily verified that if $\delta < \frac{\alpha\theta\lambda_{\min}(\boldsymbol{K})}{2\lambda_{\max}(\boldsymbol{K})}\sqrt{\frac{\lambda_{\min}(\boldsymbol{K})}{\lambda_{\max}(\boldsymbol{K})}}\eta$, then $c < b$, which implies $\chi_c \subset \chi_b$. Note, this condition satisfies the necessary condition (64) for the non-empty region since $\lambda_{\min}(\boldsymbol{K}) \le \lambda_{\max}(\boldsymbol{K})$.

For any $\boldsymbol{x}_{\mathrm{R}}$ inside $\chi_b$, using (60), we have

$$\lambda_{\min}(\boldsymbol{K})\|\boldsymbol{x}_{\mathrm{R}}\|^2 \le \mathcal{V}_{\mathrm{R}}(\boldsymbol{x}_{\mathrm{R}}) < b = \lambda_{\min}(\boldsymbol{K})\eta^2, \tag{65}$$

implying $\|\boldsymbol{x}_{\mathrm{R}}\| < \eta$. Similarly, for any $\boldsymbol{x}_{\mathrm{R}}$ on the boundary or outside of $\chi_c$, we have

$$\lambda_{\max}(\boldsymbol{K})\mu^2 = c \leq \mathcal{V}_{\mathrm{R}}(\boldsymbol{x}_{\mathrm{R}}) \leq \lambda_{\max}(\boldsymbol{K})\|\boldsymbol{x}_{\mathrm{R}}\|^2, \tag{66}$$

which implies $\|\boldsymbol{x}_{\mathrm{R}}\| \geq \mu$.

Combining (65) and (66) we can say for any $\boldsymbol{x}_{\mathrm{R}}$ outside (including the boundary) of $\chi_c$, but inside $\chi_b$, (63) holds true. For such $\boldsymbol{x}_{\mathrm{R}}$ (i.e. $\boldsymbol{x}_{\mathrm{R}} \in \chi_b \setminus \chi_c$) we have

$$\frac{d\mathcal{V}_{\mathrm{R}}}{dt} \leq -\frac{\alpha(1-\theta)\lambda_{\min}(\boldsymbol{K})}{\lambda_{\max}(\boldsymbol{K})}\mathcal{V}_{\mathrm{R}}(\boldsymbol{x}_{\mathrm{R}}) \overset{\triangle}{=} -2\gamma\mathcal{V}_{\mathrm{R}}(\boldsymbol{x}_{\mathrm{R}}), \tag{67}$$

using (60) and (63).

If the initial point (at time $t_0$) satisfies $\|\boldsymbol{x}_{\mathrm{R}}(t_0)\| < \sqrt{\frac{\lambda_{\min}(\boldsymbol{K})}{\lambda_{\max}(\boldsymbol{K})}}\eta$, then by (60),

$$b = \lambda_{\min}(\boldsymbol{K})\eta^2 > \lambda_{\max}(\boldsymbol{K})\|\boldsymbol{x}_{\mathrm{R}}(t_0)\|^2 \geq \mathcal{V}_{\mathrm{R}}(\boldsymbol{x}_{\mathrm{R}}(t_0)),$$

which implies the initial point $\boldsymbol{x}_{\mathrm{R}}(t_0)$ is inside $\chi_b$. Assuming an initial point in $\chi_b \setminus \chi_c$, and integrating (67) in time interval $[t_0, t]$, we get

$$\mathcal{V}_{\mathrm{R}}(\boldsymbol{x}_{\mathrm{R}}(t)) \leq \mathcal{V}_{\mathrm{R}}(\boldsymbol{x}_{\mathrm{R}}(t_0))e^{-2\gamma(t-t_0)}. \tag{68}$$

Hence, $\lambda_{\min}(\boldsymbol{K})\|\boldsymbol{x}_{\mathrm{R}}(t)\|^2 \leq \mathcal{V}_{\mathrm{R}}(\boldsymbol{x}_{\mathrm{R}}(t)) \leq \mathcal{V}_{\mathrm{R}}(\boldsymbol{x}_{\mathrm{R}}(t_0))e^{-2\gamma(t-t_0)} \leq \lambda_{\max}(\boldsymbol{K})\|\boldsymbol{x}_{\mathrm{R}}(t_0)\|^2 e^{-2\gamma(t-t_0)}$ as long as $\boldsymbol{x}_{\mathrm{R}}(t)$ remains outside of $\chi_c$. Since $\frac{d\mathcal{V}_{\mathrm{R}}}{dt}$ is always negative outside of $\chi_c$, any trajectory starting outside of it, must enter $\chi_c$ in finite time. Let the trajectory starting at $\boldsymbol{x}_{\mathrm{R}}(t_0)$ enters $\chi_c$ for the first time at time $t_c + t_0$. Then, we have

$$\|\boldsymbol{x}_{\mathrm{R}}(t)\| \leq \sqrt{\frac{\lambda_{\max}(\boldsymbol{K})}{\lambda_{\min}(\boldsymbol{K})}}e^{-\gamma(t-t_0)}\|\boldsymbol{x}_{\mathrm{R}}(t_0)\| = \lambda e^{-\gamma(t-t_0)}\|\boldsymbol{x}_{\mathrm{R}}(t_0)\|, \quad \forall\, t_0 \leq t < t_c + t_0. \tag{69}$$

Once a trajectory enters $\chi_c$, it cannot escape $\chi_c$ because $\frac{d\mathcal{V}_{\mathrm{R}}}{dt}$ is negative on the boundary. Therefore, all points of a trajectory after $t \geq t_c + t_0$ satisfies $\lambda_{\min}(\boldsymbol{K})\|\boldsymbol{x}_{\mathrm{R}}(t)\|^2 \leq \mathcal{V}_{\mathrm{R}}(\boldsymbol{x}_{\mathrm{R}}(t)) \leq c$, equivalently,

$$\|\boldsymbol{x}_{\mathrm{R}}(t)\| \leq \sqrt{\frac{\lambda_{\max}(\boldsymbol{K})}{\lambda_{\min}(\boldsymbol{K})}}\mu = \frac{2\delta}{\alpha\theta}\left(\frac{\lambda_{\max}(\boldsymbol{K})}{\lambda_{\min}(\boldsymbol{K})}\right)^{3/2} = \frac{2\delta}{\alpha\theta}\lambda^3, \quad \forall\, t \geq t_c + t_0. \tag{70}$$

From (67), (69) and (70), we have $\gamma = \frac{\alpha(1-\theta)\lambda_{\min}(\boldsymbol{K})}{2\lambda_{\max}(\boldsymbol{K})}$ and $\lambda = \sqrt{\frac{\lambda_{\max}(\boldsymbol{K})}{\lambda_{\min}(\boldsymbol{K})}}$. ∎

## B  Details on reaction–diffusion system experiment

### B.1  Dataset

We use FEniCS (Logg et al. (2012)), an open-source computing platform for solving PDEs using the finite element method, with Python interface to generate the dataset. For the reaction-diffusion system of (34), we generate 100 training sequences of length 50 with time step size 0.01 and 256 nodes in $\mathbb{I}$. The initial conditions and actuations of these sequences are given by

$$q(\zeta, 0) = |a| \sum_{k=0}^{4} b_k T_k(\zeta), \quad \zeta \in \mathbb{I}, \tag{71}$$

and

$$w(t_i) = 10 g_i \max_{\zeta} |q(\zeta, t_{i-1})|, \quad i = 1, 2, \cdots, 49, \tag{72}$$

where $T_k$ denotes the $k^{\mathrm{th}}$ Chebyshev polynomial of the first kind, and $a \sim \mathcal{N}(0, 1)$, $b_k, g_i \sim \mathcal{U}(-1, 1)$ are chosen randomly. Similarly, 100 sequences are generated for the test set to evaluate the prediction performance.

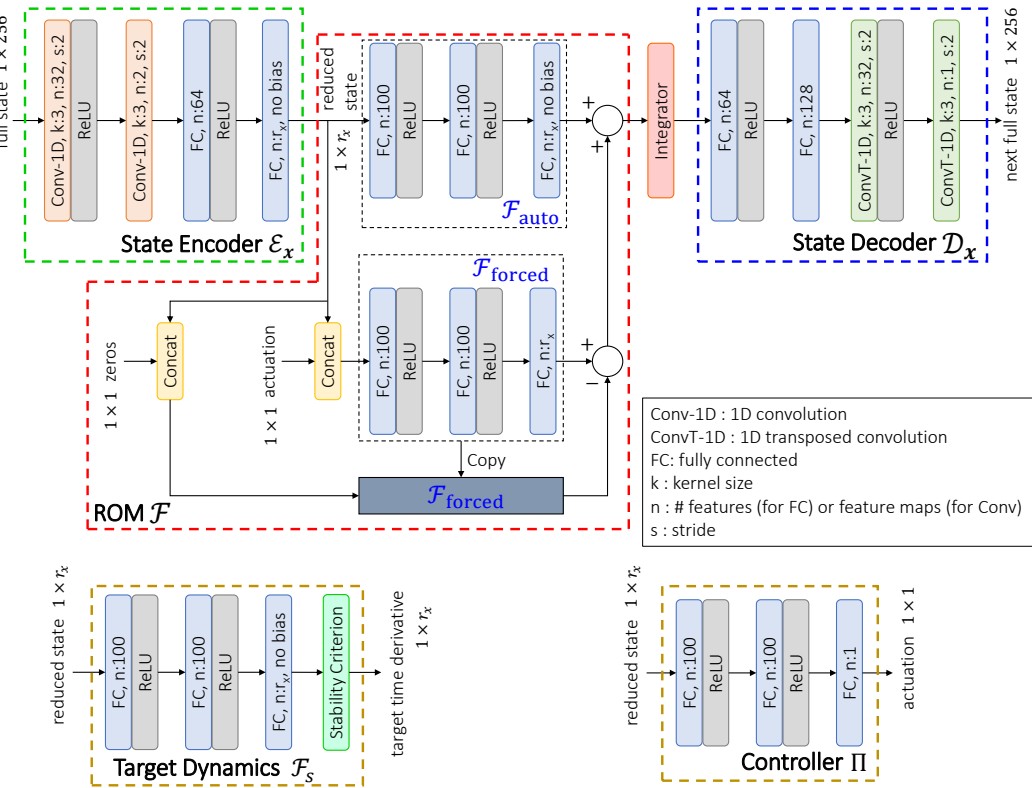

Figure 12: Architectures for all the DNN modules used in the reaction–diffusion experiment. The 'Copy' operation denotes the reuse of the same DNN block for zero and nonzero actuation. The 'Concat' operator concatenates the input features along the last dimension. Zeros are concatenated to the reduced state to evaluate the component $\mathcal{F}_{\text{forced}}(\boldsymbol{x}_{\text{R}}, \boldsymbol{0})$. The 'Integrator' performs the numerical integration for (21). The 'Stability Criterion' block implements (28).

### B.2 DNN architectures

Figure 12 shows the DNN architectures used for different modules in the reaction–diffusion experiment. The state encoder comprises 1D convolutional layers, followed by fully connected layers. The state decoder has the reversed order with convolutional layers replaced by transposed convolutional layers. The ROM is designed by breaking the function $\mathcal{F}$ into two components: $\mathcal{F}(\boldsymbol{x}_{\text{R}}, \boldsymbol{u}_{\text{R}}) = \mathcal{F}_{\text{auto}}(\boldsymbol{x}_{\text{R}}) + \mathcal{F}_{\text{forced}}(\boldsymbol{x}_{\text{R}}, \boldsymbol{u}_{\text{R}}) - \mathcal{F}_{\text{forced}}(\boldsymbol{x}_{\text{R}}, \boldsymbol{0})$. $\mathcal{F}_{\text{auto}}$ represents the autonomous dynamics that does not depend on the actuation, whereas $\mathcal{F}_{\text{forced}}$ is responsible for the impact of actuation on dynamics. The composition $\mathcal{F}_{\text{forced}}(\boldsymbol{x}_{\text{R}}, \boldsymbol{u}_{\text{R}}) - \mathcal{F}_{\text{forced}}(\boldsymbol{x}_{\text{R}}, \boldsymbol{0})$ ensures that the component responsible for learning the impact of actuation on the dynamics provides nonzero output only when the actuation is nonzero. Two multilayer perceptions (MLPs) are used to implement $\mathcal{F}_{\text{auto}}$ and $\mathcal{F}_{\text{forced}}$. This specific structure of the ROM is not crucial and a single neural network representing $\mathcal{F}(\boldsymbol{x}_{\text{R}}, \boldsymbol{u}_{\text{R}})$ works as well. However, we observe better performance in experiments when the aforementioned structure is used. The output of the ROM is integrated using a numerical integrator to get the next state. The controller is implemented using an MLP. The target dynamics is implemented using another MLP, followed by a stability criterion in the form of (28).

### B.3 Training settings

We use $r_{\boldsymbol{x}} = 5$ in the prediction task and $r_{\boldsymbol{x}} = 2$ in the control task for all the methods. All modules are implemented in PyTorch. In both of the learning phases, learning ROM and learning controller, we use the Adam optimizer with an initial learning rate of 0.001 and apply an exponential scheduler with a decay of

0.99. Modules are trained for 100 epochs in mini-batches of size 32. 10% of the training data is used for validation to choose the best set of models. For DeepROM training, we use $\beta_2 = 1$ in (22). For learning control, we use $\beta_3 = 0.2$ in (26), $\alpha = 0.2$ in (28), and $K = 0.5 I_{r_x}$ in (29). Since the learned ROMs from one training instance to another can vary, the hyperparameter pair $(\alpha, \beta_3)$ may require re-tuning accordingly.

## C  Details on vortex shedding suppression experiment

### C.1  Dataset

For the flow past a circular cylinder problem, the geometry and physical parameters of the system are taken from the DFG 2D-2 benchmark (Schäfer et al. (1996)). The geometry is shown in Figure 13. We use the blue-shaded region for observation and actuation. Following the DFG 2D-2 benchmark, we use the no-slip boundary condition of zero velocity for the walls and the cylinder boundary, zero outlet pressure, and the inflow velocity profile (at the inlet) as

$$v(\zeta, t) = \left(1.5 \frac{4\zeta_2(0.41 - \zeta_2)}{0.41^2}, 0\right), \tag{73}$$

where $\zeta_1$ and $\zeta_2$ denote the horizontal and vertical coordinates, respectively, of $\zeta$. We use kinematic viscosity $\nu = 0.002$ and density $\rho = 1$ leading to the Reynolds number $Re = 50$. The training sequence of length 5000 is generated in FEniCS with a time step size 0.001 and applying actuations

$$w(\zeta, t) = a \sum_{k=0}^{4} \left[\sin(k\pi(\zeta_1 - 0.11)/0.66) \quad \sin(k\pi\zeta_2/0.41)\right] \begin{bmatrix} b_{k,1,1} & b_{k,2,1} \\ b_{k,1,2} & b_{k,2,2} \end{bmatrix}, \quad \zeta \in \mathbb{W}, \tag{74}$$

where $a \sim \mathcal{U}(0, 1)$ and $b_{k,i,j} \sim \mathcal{U}(-1, 1), i, j = 1, 2$ are chosen randomly. Similarly, a test sequence is generated to evaluate the prediction performance. For learning control, we use the Stokes flow or creeping flow as the desired state, which can be obtained by solving the Stokes equations

$$\nu\nabla^2 v - \frac{1}{\rho}\nabla p = 0, \quad \nabla \cdot v = 0 \quad \text{in } \mathbb{I} \times \mathbb{R}^+. \tag{75}$$

For training, the flow velocity data from the observation region (blue shaded in Figure 13) are interpolated onto a rectangular uniform grid of size $32 \times 48$ so that it can be used in standard CNNs.

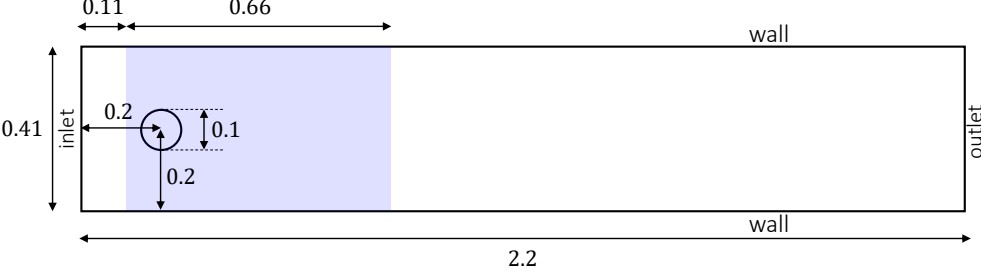

Figure 13: Geometry of the flow past a circular cylinder set-up.

### C.2  DNN architectures

Figure 14 shows the DNN architectures used for different modules in the vortex shedding control experiment. The architectures for the ROM and target dynamics are the same as in the previous example. Moreover, the state encoder and decoder have similar architectures as the previous example except for the 1D convolutions and transposed convolutions are replaced by their 2D counterparts. Here, an additional module is used: the control encoder for encoding the distributed control/actuation. It has the same architecture as the

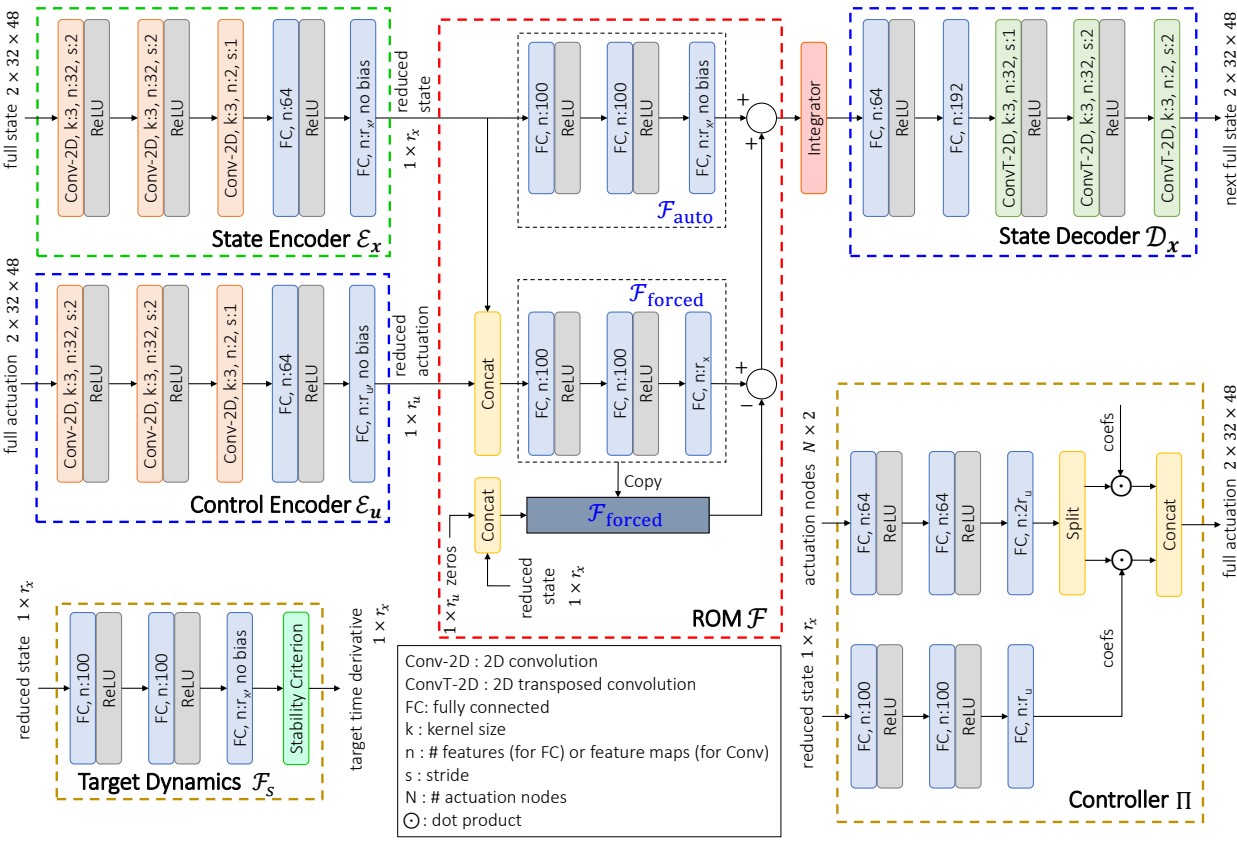

Figure 14: Architectures for all the DNN modules used in the fluid flow experiment. The 'Split' operator splits the input features into two vectors, along the last dimension. These split vectors represent the space-dependent polynomial basis associated with the horizontal and vertical components of the actuation.

state encoder. To learn the distributed actuation, we design the controller as a linear combination of space-dependent polynomial basis functions. One MLP is used to learn these space-dependent polynomial basis functions given the locations of the actuation nodes and another MLP is used to learn the corresponding coefficients. The actuation is computed as the dot product of the polynomial basis terms and the coefficient vector. We use this architecture instead of a standard convolutional one because the PDE solver takes the actuation input in a triangular mesh, not in a uniform rectangular grid. The polynomial basis architecture can be used to compute actuation in both uniform rectangular grid during training and triangular mesh during evaluation.

## C.3 Training settings

We use $r_{\boldsymbol{x}} = 5$ in both the prediction task and control task for all the methods. All modules are implemented in PyTorch. In both of the learning phases, learning ROM and learning controller, we use the Adam optimizer with an initial learning rate of 0.001 and apply an exponential scheduler with a decay of 0.99. Modules are trained for 100 epochs in mini-batches of size 32. 10% of the training data is used for validation to choose the best set of models. For DeepROM training, we use $\beta_2 = 1$ in (22). For learning control, we use $\beta_3 = 2$ in (26), $\alpha = 0.1$ in (28), and $\boldsymbol{K} = 0.5\boldsymbol{I}_{r_{\boldsymbol{x}}}$ in (29). Since the learned ROMs from one training instance to another can vary, the hyperparameter pair $(\alpha, \beta_3)$ may require re-tuning accordingly.

# D   Architecture and training details for the Deep Koopman model

For the encoder and decoder of the Deep Koopman model, we use the same architectures as our state encoder and state decoder. As mentioned in section 5.1, we consider both the system and input matrices of the ROM to be fixed during operation, in contrast to the original method proposed by Morton et al. (2018). Therefore, during training, these matrices are treated as trainable global parameters. Similar to Morton et al. (2018), the input matrix is optimized by gradient descent during training along with the encoder-decoder parameters, whereas the system matrix is obtained using linear least-squares regression. The datasets are divided into staggered 32-step sequences for training, and the model is trained by generating recursive predictions over 32 steps following Morton et al. (2018). We train the model using the Adam optimizer with an initial learning rate of 0.001 and an exponential decay of 0.99 for 200 epochs in mini-batches of size 8. 10% of the training data is used for validation to choose the best set of models.

As mentioned in 5.3.3, we utilize a low-dimensional representation of the distributed actuation for Deep Koopman + LQR, instead of directly estimating the high-dimensional actuation. The distributed actuation is represented as a linear combination of the same space-dependent sinusoidal basis functions used for dataset generation, which are given by (74). The controller is designed to estimate the coefficients $b_{k,i,j}; i,j = 1,2; 0 \leq k \leq 4$.

