# OpenReview forum: "Autoencoding Reduced Order Models for Control through the Lens of Dynamic Mode Decomposition"
_TMLR — Rejected by TMLR_

### Review · Reviewer_arzY · 2023-05-09

**Summary Of Contributions:**

The authors present a new approach for designing controllers for dynamical systems by combining dynamic mode decomposition and autoencoders. The approach, first  involves formulating an optimization objective for learning a reduced order model using a linear autoencoder architecture that closely resembles the solution obtained by dynamic mode decomposition with control. It is then extended to a deep autoencoder architecture to learn a nonlinear reduced order model, which is used to design a controller utilizing stability-constrained deep neural networks. The authors note that the method does not require knowledge of the governing equations of the system and learns the model and controller solely from time series data of observations and actuations.

**Audience:**

Yes

**Claims And Evidence:**

Yes

**Requested Changes:**

See above

**Strengths And Weaknesses:**

I have the following comments for the authors.

1. The authors claim that the use of their so-called LAROM “can be used to approximate modal decomposition for high-dimensional systems where computing SVD is expensive”. I find this claim a bit strong - a gradient-descent based procedure to compute the modal decomposition, in my experience, can never match, say, randomized or streaming techniques for the SVD which have been studied exceptionally thoroughly in the past. Can the authors comment on this?

2. What was the computational benefit of going through the trouble to fit the nonlinear autoencoders first, and the control encoder next in comparison with DMDc. Can the authors provide comparisons for training and inference time between the pair? This will be important for whether their procedure is overall beneficial for this problem.

3. Perhaps I missed this - but the authors use a neural ODE based training technique to train for dynamics in the latent space. I was not clear about the motivation behind this. Since the snapshots are presumably equispaced, why not use a regular apriori first-order discretization of the dynamics and train the network. This would also make training of the control faster would it not? They mention that the subsection of learning control would make this clear - and I presume it is connected with their F_s target dynamics in Equation 29 - but it is not very apparent.

4. The authors describe the formulation of LAROM to be constrained for a similar structure to DMDc, but it remains a bit ambiguous about the semi-orthogonality constraint and lack of optimality for DMD. In particular, in 5.1.1 there is a constraint for orthogonality imposed on what is understood to be the ROM similar to DMD, but orthogonal projections are a product of the SVD of the full rank dataset to define a projection and then subsequently computing DMD on the reduced representation. Although a relatively minor detail, DMD modes themselves are not orthogonal nor are they ranked by any optimality inherently. There is a comparison between leading DMD modes (presented in figures 4, 5, 7, 8, and 9) but no discussion on whether these are ranked by energy content of the POD modes (in the case of DMD). For autoencoders there is also a lack of optimality and orthogonality inherently, and so a direct comparison between leading modes of the LAROM and DMDc is misleading if there is no discussion on the lack of optimality of both of these methods.

5. Finally, I would like to point out that the discussion of exponential stability in Theorem 4.2.1, while guaranteeing exponential stability of dynamics in the latent space of the nonlinear ROM, does not really mean anything with regard to the control of the original system. Just because dynamics are constrained in the latent space - does not imply any theoretical guarantee of the dynamics in the original space. Can the authors comment on what makes this theorem useful for actually controlling a high-dimensional system?

---

> ### Author Response · Authors · 2023-06-19
> **Response to Reviewer arzY**
>
> We thank the reviewer for their valuable time and feedback. We are revising the paper based on the comments/suggestions provided.
> Here we address the reviewer’s comments.
>
> 1. We agree with the reviewer that the randomized or streaming techniques are more accurate and efficient than gradient descent-based procedures to compute SVD or modal decomposition. The paper does NOT aim to provide an alternative method to compute modal decomposition. The similarity in the modal decomposition with DMDc and LAROM is shown to establish a connection between the two methods which builds the foundation for a deep learning-based approach.  The primary purpose of the gradient descent-based framework is to incorporate DNN-based nonlinear models. We will remove the misleading statement from the revised paper.
>
> 2. In this paper, the purpose of fitting nonlinear autoencoders is NOT to achieve computational benefits over DMDc. The goal of this paper is to investigate if nonlinear DNN-based models can improve the prediction and control performance over linear models. As shown in experiments, the proposed method has better prediction accuracy than the linear model over a longer period (in time). Moreover, the proposed method requires less control effort than its linear counterpart for the first experiment. However, the benefit in control performance of the second experiment is not significant.
> We do NOT expect the DNN-based models to have less training and inference time compared to the linear models. The use of deep autoencoders for dimensionality reduction in existing literature highlights the benefits of a better approximation accuracy over the computational benefits. This paper shares the same purpose for using nonlinear ROMs.
>
> 3. Yes, the reason behind using neural ODE or any differentiable numerical integrator is connected to the learning of the control. The output of the stability-constrained DNN is given by the RHS of eqn. (29) which represents the vector field $\mathcal{F}_s$ of the target dynamics. This $\mathcal{F}_s$ is used in the MSE objective of eqn. (26) to train the controller. According to (26), we need the vector field of the learned ROM to train the controller. Therefore, we formulate the ROM to be in continuous-time form so that it provides the vector field $\mathcal{F}(x_r, u_r)$ of the underlying dynamics. This vector field $\mathcal{F}$ is then integrated using a differentiable numerical integrator to get the next state for training the autoencoder-based ROM using eqn. (23). In our example test cases, we used equispaced samples for training. However, this is not a requirement for the proposed method and irregular time samples can be used straightforwardly as we are utilizing numerical integrator or neural ODE anyway. In cases where only the prediction model is of interest and control learning is not required, a first-order discretization of the dynamics should be used for faster training of the ROM. We apologize for not stating this clearly in the paper and we will revise the paper accordingly.
>
> 4. The purpose of the semi-orthogonality constraint used in section 5.1.1, is to impose a structure on the encoding matrix $\mathcal{E}_x$. Similar to SVD-based dimension reduction methods, we intend to obtain an encoding matrix that is semi-orthogonal so that the encoded features are linearly independent (like the POD modes in the case of DMD).  However, we only include a soft constraint for semi-orthogonality in the objective function optimized using gradient descent. Therefore, the latent modes (not the dynamic modes) from LAROM will not be exactly orthogonal, though their correlations are minimized. Moreover, we observe that the latent modes from LAROM can be recovered with high accuracy if we project them onto the subspace spanned by the POD modes (from DMDc) and reconstruct them back. This means that the two subspaces associated with latent modes from LAROM and the POD modes from DMDc are close. We are open to including these relations among the POD modes from DMDc and latent modes from LAROM in the revised paper if the reviewer feels that will clear the ambiguity.
> Unlike the POD modes, the ranking of the LAROM modes is arbitrary as we did not include any ranking constraint in the optimization objective. However, we verified the correspondence between the DMDc and LAROM modes based on the eigenvalues of the system/state matrices of the ROMs obtained from the two methods.

---

> > ### Author Response · Authors · 2023-06-19
> > **Continued from the previous response**
> >
> > 5. Yes, guaranteeing the stability of the reduced-order dynamics does not provide any theoretical guarantee of the dynamics in the original space. On the other hand, if the reduced-order dynamics is NOT stable (at the desired point) under the proposed control policy, then the original system surely would not be stable. Hence, Theorem 4.2.1 serves as a necessary (but NOT sufficient) condition for the stability of the original system. We will add this discussion to the revised version for clarity.

---

### Review · Reviewer_dkKx · 2023-05-29

**Summary Of Contributions:**

Inspired by previous works of learning a simpler model of a high-dimensional dynamical system via dynamic mode decomposition (and its extension for controlled dynamical systems), the authors demonstrate that a similar solution can be obtained by optimizing a particular loss function. The authors extended this insight to deep neural networks, allowing for a non-linear encoder, decoder, and dynamics. With the approximation in hand, the system is controlled by learning a feedback controller that tries to have the learned model match a hypothesized, stable non-linear dynamical system. Excitedly, the authors prove that such an approach guarantees that the controller applied to the model will ensure that the system's state is always near the target state.

**Audience:**

Yes

**Broader Impact Concerns:**

I don't see any ethical implications.

**Claims And Evidence:**

Yes

**Requested Changes:**

Below I will list changes and also questions that I have.

1) Figure 1b seems to imply that one optimizes a linear autoencoder and feeds it into a non-linear autoencoder which is different from what is happening. The figure should be changed to demonstrate that one can use either a LAROM or a deep-learning-based one. (*critical*)
2) The paper would read better if the semi-orthogonality constraint introduced in the experiments section was discussed in the LAROM methods section. (*critical*)
3) Since the authors assume that the true system is Lyapunov stable, why not enforce the ROM model *\mathcal{F}* to be Lyapunov stable via the construction demonstrated in equation 29? A priori, this seems like it would avoid the need to learn a separate hypothesized neural network. (*not critical*)
4) I really appreciate the diagrams in the appendix demonstrating the architecture, but some design choices aren't clear. For instance, why is a zero concatenated to the reduced state? Some details on the design choices of the networks would be incredibly enlightening. (*critical*)

**Strengths And Weaknesses:**

# Strengths

In general, I am a big fan of this approach. It is technically sound and easy to understand. While I think the delivery could be cleaned up, the paper is relatively easy to read. Moreover, I *love* the insights provided by the empirical section of this paper: 1) better prediction accuracy of the dynamics doesn't always translate to better controller performance, and 2) the deep-learning-based approach is not a silver bullet for every problem.

# Weaknesses

The two most significant weaknesses, in my opinion, are the writing and the experiments section. In terms of writing, I think the heavy use of acronyms obfuscates the method extensively. Moreover, it isn't clear what the difference is between a model-order reduction and a reduced-order model; in my reading, the two terms seemed to be used interchangeably.

I think it would be great to have a simple deep model-based/model-free RL baseline operating directly on the raw states for the experiments section. This would serve two purposes: 1) demonstrate how long it would take a baseline deep RL algorithm to do the task and 2) demonstrate the pro of learning a reduced order model.

---

> ### Author Response · Authors · 2023-06-19
> **Response to Reviewer dkKx**
>
> We thank the reviewer for their valuable time and feedback. We are pleased that the reviewer liked our approach and found the empirical section insightful.  We are revising the paper based on the comments/suggestions provided. Here we address the reviewer’s comments.
>
> 1. We will replace the acronyms with clear phrases wherever possible in the revised version. By ‘reduced-order model’, we refer to the lower dimensional model itself while the term ‘model-order reduction’ is used to describe the process of dimensionality reduction. We will replace the term ‘model-order reduction’ with a more clear phrase (e.g. reduced-order modeling) in the revised version.
>
> 2. As we mentioned in the Related Work section of the paper, most of the existing RL-based systems use system-specific rewards to train the controller for complex systems like fluid flow. In this paper, we investigate a generic scenario without specifically focusing on a particular system. Training an RL policy with such a setting directly on the raw states (by comparing the difference between high-dimensional states) is extremely challenging.  We agree that it would be interesting to investigate the advantages of learning a reduced order model over RL-based methods on the original states in terms of training time, sample complexity, and performance. While we focus on the feasibility of control learning through deep reduced-order models and their relations with existing reduced-order model-based methods in this paper, we aim to empirically investigate the advantage of such methods over RL-based techniques on original states in a future work.
>
> 3. We agree that current Figure 1b can be confusing to the reader. We will modify it in the revised version.
>
> 4. We will move the discussion on the semi-orthogonality constraint from the experiment section to the LAROM method section.
>
> 5. We do NOT assume that the true system is Lyapunov stable without any control input acting on it. We assume that the true system is stabilizable with an appropriate control policy. So, we need a control policy to make it stable. The ROM learned in the first phase (the model learning phase) is not constrained to be Lyapunov stable and is a generic reduced-order representation of the true system. The neural network for the hypothesized stable dynamics is required to guide the learning of an appropriate control for the ROM (which is not stable in general) such that the closed-loop system becomes stable. This hypothesis acts as a target for the closed-loop system to match.
>
> 6. We split the ROM into an autonomous part ($\mathcal{F}_1$) and a forced/controlled part ($\mathcal{F}_2$) as follows:
> \begin{align}
>     \frac{dx_\mathrm{R}}{dt} &=\mathcal{F}\left(x_\mathrm{R},u_\mathrm{R}\right) …………………………………(1) \newline
> &=\mathcal{F}\left(x_\mathrm{R},\ 0\right)+\ (\mathcal{F}\left(x_\mathrm{R},u_\mathrm{R}\right)\ -\ \mathcal{F}\left(x_\mathrm{R},\ 0\right)) \newline
> &= \mathcal{F}_1\left(x_\mathrm{R}\right)+\ \mathcal{F}_2\left(x_\mathrm{R},u_\mathrm{R}\right)
> \end{align}
> We use two neural networks to represent $\mathcal{F}_1$ and $\mathcal{F}_2$. Now, to ensure that $\mathcal{F}_2$ provides nonzero output only when the actuation $ u_\mathrm{R}$ is nonzero, we structure $\mathcal{F}_2\$ as
> \begin{align}
> {\mathcal{F}_2\left(x_\mathrm{R},u_\mathrm{R}\right)\ =\ \mathcal{F}}_3\left(x_\mathrm{R},u_\mathrm{R}\right)\ -\ \mathcal{F}_3\left(x_\mathrm{R},\ 0\right),
> \end{align}
> where $\mathcal{F}_3$ acts as the basic block for designing $\mathcal{F}_2$. Hence, zero is concatenated to the reduced state to evaluate the component $\mathcal{F}_3\left(x_\mathrm{R},\ 0\right)$.
> This specific structure is not necessary. Just a single neural network representing eqn. (1) works as well. However, we observe better performance in experiments when the aforementioned split representation is used. We think this representation helps in learning the behavior of the system with and without actuation. A better representation of the effect of actuation leads to better control policy.

---

### Review · Reviewer_d42b · 2023-06-11

**Summary Of Contributions:**

The author claimed to have four contributions:
1. For controlled dynamical systems, we show that an objective function can be formulated in a linear autoencoding configuration and optimized by gradient descent such that the corresponding linear ROM closely resembles the ROM obtained using the DMDc algorithm.
2. We extend the linear autoencoding configuration to a deep autoencoding configuration to learn a DNN-based nonlinear ROM.
3. We analytically show that a DNN controller can be trained such that the closed-loop trajectories of the learned ROM remain ultimately bounded.
4. We empirically show the similarity of the linear autoencoding ROM (LAROM) with DMDc and evaluate the prediction performance of DeepROM and control performance of DeepROC in experiments with high-dimensional systems, including suppressing vortex shedding in fluid flow.

My interpretation of the claimed contributions:
1. An analysis showing linear autoencoder and DMDc algorithm learns a similar solution (dynamic system).
2. A method (loss) to train ROM using a deep neural network. In my view, the proposed method is **NOT** new and had been used by many prior works such as [1].
3. An analysis showing the learned DNN controller can achieve ultimately bounded.
4. Empirical results to support (1) linear autoencoder and DMDc learning similar things; (2) DeepROM does well in high-dimensional systems.

[1] Li, Yunzhu, et al. "Learning compositional koopman operators for model-based control." arXiv preprint arXiv:1910.08264 (2019).

**Audience:**

No

**Broader Impact Concerns:**

This paper does not have a Broader Impact section.

**Claims And Evidence:**

No

**Requested Changes:**

The requested changes are listed in the weakness section.

**Strengths And Weaknesses:**

Pros
---
1. The author provides an analysis revealing the similarity between DMDc and linear autoencoder.
2. Experiments show the proposed DeepROM outperforms a simple baseline DMDc on simple PDE systems.
3. The author shows that the proposed method provides stability.

Cons
---
**Positioning.**
After reading the related work section, as a reader, I cannot get the position of this paper in the big picture of the area. The authors seem just name prior works without drawing connections between the works, especially the relationship to their work. Basically, the first paragraph is about using DL for dynamics modeling, the second paragraph is about using DL to do control, and the third paragraph is about using DL model predictive control. Those related works are too general, could even be attached to any paper in this field, and do not really provide a context, a motivation for this work. I will suggest the author focus on demonstrating the **delta** between their work and prior works in the related work section.

**Clarity.**
The paper's writing is not clear.
1. Figure 1: The caption is too short to provide enough explanation of the figure. For example, the sharp corner boxes and the rounded corner boxes should be different meanings but are not articulated in the caption. The usage of the color is also not explained.

2. Section 3.2: The section aims to introduce the concept of (locally) stabilizable, globally stabilizable, exponentially stable. The organization of the paragraphs is poor. The paragraphs for different concepts are mixed together which makes it confusing. The statements are also not precise due to the improper usage of brackets. For example, it says "We assume that the system  ....  is stabilizable in the sense of the aforementioned definition and criteria, i.e., there exists a continuously differentiable function V and a Lipschitz continuous control law π such that criteria (3) and (4) **(and possibly (5) and (7) as well)** are conformed." It seems (3)(4) are about locally stabilizable and (5)(7) are about globally stabilizable or exponentially stable. From the author's statement, I cannot understand that the author is assuming the system to be locally stabilizable or globally stabilizable or exponentially stable. What are the necessary conditions for their method to work.

3. Section 4.1: This is a long section composed of many pieces: (1) analysis for a fixed encoder, (2) connection between linear autoencoder's solution and the DMDc's solution; (3) change for linear autoencoder to deep autoencoder by replacing linear layer with deep neural networks. The transition between the pieces is not smooth. Sometimes motivation is missing. For example, the author says "Accordingly, we propose to minimize ..." and then proposes equation (19). As a reader, I have no idea what motivates the author to propose (19). Further, equation (23) is similar to (19) while the difference seems to be (19) is discrete timestep while (23) is continuous time. I am not sure why moving from linear autoencoder to deep autoencoder also causes a jump from discrete to continuous.

**Experiments.**
1. Choice of tasks: the authors directly propose to evaluate their methods on "spatiotemporal PDE-driven systems" without validating why they are the right tasks to verify their method. Can the author provide more insights into why choosing those tasks and specific datasets? Are they challenging? Are they standard benchmarks used by many prior works?
2. Lack of baselines: the authors only compare their method with simple baselines LAROM and DMDc. As they listed in the related work section, there should be many prior works that use DL to do dynamic modeling and control. Can the author include more recent and more advanced baselines such as [2]?

[2] Bethany Lusch, J Nathan Kutz, and Steven L Brunton. Deep learning for universal linear embeddings of nonlinear dynamics. Nature communications, 9(1):4950, 2018.

---

> ### Author Response · Authors · 2023-06-22
> **Response to Reviewer d42b**
>
> We thank the reviewer for their valuable time and feedback. We are revising the paper based on the comments/suggestions provided. Here we address the reviewer’s comments.
>
> 1. We are NOT claiming that the idea of learning a DNN-based ROM is new. We agree that there are several recent works that use similar autoencoding models to learn dynamics. We clearly mentioned in the introduction that “the architecture with DNN components, DeepROM, closely resembles the deep autoencoding architectures used in recent literature for the prediction and control of dynamical systems”.
> Our contribution to learning a DNN-based ROM is to connect this emerging trend to the existing established method of dynamic mode decomposition for controlled dynamical systems. It is possible to design and train the autoencoding ROMs in multiple ways, as shown in some recent literature. Our specific method is formulated to establish the connection between the two techniques through a linear autoencoding representation of DMDc.\
> \
> As the reviewer pointed out, some recent work like [1] used the learned ROM for model-based control. However, these methods, including [1], constrain the dynamic model to be linear, following Koopman theory. These linear ROMs are then used with conventional control methods to design the controllers. Most of the works use MPC to design controllers as the linear models generally work well within a short time window and are needed to be updated with online observations during operation. Accordingly, the control policy is optimized online using the updated dynamic model. In contrast, we investigate if a nonlinear ROM provides a more accurate prediction over a longer time window so that an offline control learning method can be used.
>
> 2. We will revise the Related Work section as suggested to highlight the difference between our work and prior works.
>
> 3. We will modify Figure 1 and add details to the caption to explain it better.
>
> 4. We agree that Section 3.2 is confusing to the readers with many different concepts introduced together and some of the information is less relevant to the content of the rest of the paper. We will revise this section to organize it better in the context of our method and remove any unnecessary information. We assume that the underlying system is **locally stabilizable** as we show that the closed-loop trajectory is bounded only when the initial point is within a certain region. We included a discussion on exponential stability as the hypothesized target dynamics is designed to be exponentially stable at the origin. Such choice of the hypothesized target dynamics is essential to ensure the boundedness of the closed-loop ROM trajectories, as shown in Theorem 4.2.1.
>
> 5. We are considering breaking section 4.1 into smaller subsections as suggested to avoid unexpected transitions from one piece to the next.
> The similarity between DMDc and LAROM is shown assuming a fixed encoder consisting of a subset of eigenvectors of the state covariance matrix $YY^\top$. $L_{pred}$ alone does not impose any constraints to learn an encoder with such properties. To encourage such an encoder, we add another loss term $L_{recon}$ (18) with the prediction loss $L_{pred}$ (13) to get the overall loss of (19). This choice of $L_{recon}$ is motivated by the work of Baldi & Hornik (1989) on the similarity between PCA and linear autoencoder. They showed that all the critical points of $L_{recon}$ correspond to projections onto subspaces associated with subsets of eigenvectors of the state covariance matrix. Therefore, optimization of $L_{recon}$ facilitates the selection of the desired encoder. We will add this discussion to the revised version to clarify the motivation.
> We use a discrete-time formulation for the linear autoencoder to establish the similarity between DMDc and LAROM since DMDc uses a discrete-time formulation. The same can be used for the deep autoencoder as well. However, we use a continuous-time formulation for the deep model because the control learning algorithm requires such formulation. According to eqn. (26), we need the vector field of the learned ROM for the training of the controller. Hence, the choice of this specific control learning algorithm is the sole reason for using a continuous-time formulation of DeepROM. If the model is used only for prediction purposes or a different control algorithm is used that is compatible for the discrete-time model, then a discrete-time DeepROM would be a suitable choice. We will add this discussion to the revised paper before jumping from discrete to continuous.

---

> > ### Author Response · Authors · 2023-06-22
> > **Continued from the previous response**
> >
> > 6. One of the primary applications of reduced-order modeling lies in comprehending the behavior of complex physical processes which are typically characterized by systems of PDEs. Since spatiotemporal PDE-driven systems are infinite-dimensional in their continuous form and high-dimensional when discretized, they are a natural choice for evaluating our method. The specific datasets considered in the paper have been previously employed in other works. For example, the reaction-diffusion example is used to evaluate nonlinear controllers designed from reduced-order state space representation [a]. Similar systems are used for modeling problems in 1D [b] and 2D [c].  The fluid flow system is a widely used benchmark [d] and is of great importance for many engineering applications. Many prior works on deep learning-based modeling and control have used this system for evaluation [e-j]. We will add this discussion in the empirical section of the revised paper.\
> > \
> > a. Dante Kalise and Karl Kunisch. Polynomial approximation of high-dimensional Hamilton–Jacobi-Bellman equations and applications to feedback control of semilinear parabolic PDEs. SIAM Journal on Scientific Computing, 2018
> > b. Maziar Raissi et al., "Physics-informed neural networks: A deep learning framework for solving forward and inverse problems involving nonlinear partial differential equations." Journal of Computational Physics, 2019
> > c. Angran Li et al. "Reaction-diffusion system prediction based on convolutional neural network." Scientific Reports, 2020
> > d. Michael Schäfer et al. Benchmark computations of laminar flow around a cylinder. Springer, 1996
> > e. Hamidreza Eivazi et al. "Deep neural networks for nonlinear model order reduction of unsteady flows." Physics of Fluids, 2020
> > f. N Benjamin Erichson et al. Physics-informed autoencoders for Lyapunov-stable fluid flow prediction. arXiv preprint arXiv:1905.10866, 2019
> > g. Jean Rabault et al., Artificial neural networks trained through deep reinforcement learning discover control strategies for active flow control. Journal of fluid mechanics, 2019
> > h. Hongwei Tang, Jean Rabault, Alexander Kuhnle, Yan Wang, and Tongguang Wang. Robust active flow control over a range of Reynolds numbers using an artificial neural network trained through deep reinforcement learning. Physics of Fluids, 2020. \
> > i.  Katharina Bieker et al. Deep model predictive flow control with limited sensor data and online learning. Theoretical and computational fluid dynamics, 2020.\
> > j. Jeremy Morton et al. Deep dynamical modeling and control of unsteady fluid flows. NeurIPS 2018.
> >
> > 7. Only autonomous systems (systems with no control input) are considered in [2]. Therefore, the method from [2] cannot be straightforwardly applied to the controlled dynamical systems we considered. However, we are working on some other prior work that uses a method similar to [2] for controlled dynamical systems. we are aiming to incorporate that as a baseline.

---

### Review · Reviewer_aTBX · 2023-06-11

**Summary Of Contributions:**

The paper aims to solve the problem of controlling a dynamical system by projecting observations to low-dimensional latent state space, solving the control problem there, and projecting back to the observation space in an auto-encoder fashion.The paper adopts ideas from three well-established methods: i) Dynamic Mode Decomposition, ii) Reduced Order Systems, iii) Auto-Encoder Networks. The paper introduces an analysis of Lyapunov stability properties of the plain linear version of the system and then extends it to the nonlinear case later. The paper reports results on a number of use cases derived from nonlinear physical systems literature.

**Audience:**

No

**Broader Impact Concerns:**

The paper does not have a broad impact section.

**Claims And Evidence:**

No

**Requested Changes:**

The paper looks far away from the level of maturity and originality expected from a journal publication. It requires a thorough modification in all aspects including hypothesis creation, its analysis, and execution. More precisely a potential revision or resubmission of the paper should: i) single out a main hypothesis, ii) sharpen the novelty claims, iii) simplify the storyline, iv) address the state of the art and numerically compare against them including recent advances in model-based reinforcement learning, v) report results on high-dimensional state spaces.

**Strengths And Weaknesses:**

Strengths:
- The paper provides a theoretical stability analysis of its proposed solution.

Weaknesses:
 - The main hypothesis of the paper is not clear. If it is an auto-encoder based neural architecture for the control of dynamical systems on a latent state space, there are plenty of studies addressing the same problem. For instance:

[1] D. Hafner et al., “Dream to control: Learning behaviors by latent imagination”, ICLR, 2010

 - Application of a Lyapunov stability analysis to ROS by DMD is a rather straightforward allocation of few textbook methods. Its linear extension may be a decent contribution, but I would not count it as a solid contribution.
- The paper points many times to high-dimensional observation spaces especially when motivating the auto-encoder architecture, however it ends up with reporting experiments in environments with dramatically small dimensionality. The state of the art in latent dynamics modeling and control is much ahead of this, as can also be seen in the environments used by the Dreamer paper [1].

---

> ### Author Response · Authors · 2023-06-25
> **Response to Reviewer aTBX**
>
> We thank the reviewer for their valuable time and feedback. Here we address the reviewer’s comments.
>
> 1. While we acknowledge that there have been previous studies on autoencoder-based neural architectures for the control of dynamical systems on a latent state space, it is important to note that our intention was not to claim novelty in this general idea. Instead, our paper aims to establish a connection between a more traditional approach and the emerging deep learning-based methods for the control of dynamical systems.\
>  We understand that there are existing studies like [1] that address control of dynamical systems on a latent state space. However, it is crucial to consider that the objectives and types of systems in our paper and [1] are significantly different, and hence, a direct comparison may not be appropriate. Similar RL approaches as [1] are also used in some prior works for the types of systems we used in our paper. Our related work section discusses the differences with those methods. \
> While RL-based methods have demonstrated remarkable success in solving complex problems, it is important to acknowledge the challenges associated with them as well. Our work, on the other hand, investigates the learning of control for dynamical systems from a more traditional perspective and presents a method capable of learning from fixed, relatively small datasets.
>
> 2. We would like to clarify a misunderstanding regarding the application of a Lyapunov stability analysis to ROS by DMD. It seems there may be confusion in the interpretation of our work. To clarify, we did not employ a stability analysis to ROS using DMD. The stability analysis we presented in our paper is specifically applied to the deep learning-based control method, and it is not related to DMD in any way.
>
> 3. We would like to highlight that the dimensions of our second experiment are 32x48x2, which are indeed smaller than the environments described in [1], which have dimensions of 64x64x3. However, we do not think the first number should be regarded as dramatically small compared to the latter in terms of dimensionality.
>
> 4. We appreciate the reviewer's suggestions regarding the requested changes. However, we would like to inform them that a comprehensive modification of all aspects of the paper is not feasible at this stage. Nonetheless, we have made substantial changes in the revised version to improve the paper and we hope it addresses some of the concerns raised.

---

### Decision · Action_Editors · 2023-08-22

**Recommendation:** Reject

**Comment:**

This paper presents an approach for designing controllers for high-dimensional dynamical systems that draws connections between dynamic mode decomposition, reduced-order systems, and deep autoencoders. It presents an empirical analysis that supports the utility of the approach and explores a number of applications. After the discussion period, reviewers were not convinced that this paper provided clear and convincing evidence in favor of their approach, and as such the paper is not a good candidate for publication at TMLR at this time. To improve the manuscript, the authors are encouraged to focus on the clarity of exposition as well as improving the framing of their contributions in relation to prior work.

**Audience:**

The TMLR audience would have many readers interested in the topic of this paper.

**Claims And Evidence:**

While some reviewers appreciated the analysis and one was even a "big fan of this approach," several reviewers brought up concerns related to the clarity of the claims and evidence. In particular, the connections to previous work are not clear enough, as even after revisions some reviewers feel that it remained unclear whether the approach is sufficiently distinct from prior work. Additionally, there are lingering concerns about the clarity of exposition, not just regarding prior work but for the paper in general. Among the important criteria for acceptance to TMLR are clear and unambiguous claims that are supported by evidence -- to meet this bar, the paper should be revised to better highlight the main claims, their relation to prior work, and how the derived results support them.

**Resubmission Of Major Revision:**

The authors may consider submitting a major revision at a later time.